:ᐧᑫᬧᔿ PLOS | ONE

# Genome wide genetic dissection of wheat quality and yield related traits and their relationship with grain shape and size traits in an elite × non-adapted bread wheat cross

Ajay Kumar [1]*, Eder E. Mantovani[1], Senay Simsek[1], Shalu Jain[2], Elias M. Elias[1], Mohamed Mergoum[1]¤*

1 Department of Plant Sciences, North Dakota State University, Fargo, ND, United States of America,
2 Department of Plant Pathology, North Dakota State University, Fargo, ND, United States of America

¤ Current address: Department of Crop and Soil Sciences, Griffin, GA, United States of America
* Ajay.kumar.2@ndsu.edu(AK); mmergoum@uga.edu(MM)

## Abstract

The genetic gain in yield and quality are two major targets of wheat breeding programs around the world. In this study, a high density genetic map consisting of 10,172 SNP markers identified a total of 43 genomic regions associated with three quality traits, three yield traits and two agronomic traits in hard red spring wheat (HRSW). When compared with six grain shape and size traits, the quality traits showed mostly independent genetic control (~18% common loci), while the yield traits showed moderate association (~53% common loci). Association of genomic regions for grain area (GA) and thousand-grain weight (TGW), with yield suggests that targeting an increase in GA may help enhancing wheat yield through an increase in TGW. Flour extraction (FE), although has a weak positive phenotypic association with grain shape and size, they do not share any common genetic loci. A major contributor to plant height was the *Rht8* locus and the reduced height allele was associated with significant increase in grains per spike (GPS) and FE, and decrease in number of spikes per square meter and test weight. Stable loci were identified for almost all the traits. However, we could not find any QTL in the region of major known genes like *GPC-B1*, *Ha*, *Rht-1*, and *Ppd-1*. Epistasis also played an important role in the genetics of majority of the traits. In addition to enhancing our knowledge about the association of wheat quality and yield with grain shape and size, this study provides novel loci, genetic information and pre-breeding material (combining positive alleles from both parents) to enhance the cultivated gene pool in wheat germplasm. These resources are valuable in facilitating molecular breeding for wheat quality and yield improvement.

## Introduction

Wheat (*Triticum aestivum* L.) is one of the major food crops of the world and has an important role to play in achieving the food security. Therefore, enhancing grain yield (GY) and

**Data Availability Statement:** All data are provided as Supporting Information.

**Funding:** The authors received no specific funding for this work.

**Competing interests:** The authors have declared that no competing interests exist.

**Abbreviations:** CIM, Composite interval mapping; DH, Dates to heading; FE, Flour extraction; GA, Grain area; GH, Grain hardness; GL, Grain length; GLWR, Grain length width ratio; GPC, Grain protein content; GPS, Grains per spike; GW, Grain width; GY, Grain yield; HRSW, Hard red spring wheat; PH, Plant height; PVE, Phenotypic variation explained; QTL, Quantitative trait locus/loci; RIL, Recombinant inbred line; SNP, Single nucleotide polymorphism; SPMS, Number of spikes per square meter; TGW, Thousand grain weight; TW, Test weight.

improving resistance to biotic and abiotic stresses is always a target for genetic improvement in this crop. In recent times, due to demands of high quality wheat in domestic and international markets, end use quality has also become one of the major target of various wheat breeding programs around the world. Hard red spring wheat (HRSW), grown primarily in the northern plains of the United States, is a specialty wheat with high protein content and superior gluten quality. Some of the world's best baked products are made from HRSW. Hard red spring wheat is also used extensively around the world as a blending wheat to increase the gluten strength and protein quantity in a batch of flour. The resulting flour has dough with improved mixing characteristics and water absorption, which can be used to make different type of bread products and noodles.

The wheat quality or end-use quality traits are mostly quantitatively inherited and controlled by a large number of QTL/genes and gene networks that are greatly influenced by environmental conditions [1–4]. More importantly, the direct estimation of several quality traits (e.g. bread making quality), through full-scale tests is expensive, time consuming and requires a large amount of grain, which is usually not available in early generation breeding lines. These limitations generally result in screening only limited number of advanced "candidate" lines. In some cases, simple indirect tests (e.g. SDS sedimentation volume for bread making quality) are used in earlier generations. However, several studies showed a week relationship between indirect test parameters and the final test scores [5–7]. If a line with desirable quality parameters could be identified in early generations, this would allow us to 1) discard undesirable lines in early generations, and 2) evaluate a larger number of populations or greater number of lines within each population at the earlier generations (head-row stage). This will help save tremendous amount of resources in the breeding program and also enhance our chances of recovering improved lines. Modern genomic tools offer a great opportunity to predict the end use performance and other quality characteristics of wheat lines in early generations without conducting the direct tests for quality [4, 8, 9]. Moreover, the availability of important genomic resources like whole genome sequence information for wheat [10] and improved gene editing technologies in recent time [11], also offer new opportunities to manipulate important individual genes to improve phenotypic performance. Therefore, to fully exploit the available genomic resources to achieve genetic gain in short time, it is important to have the knowledge about the genetics of the target traits. The identification of the QTL/genes influencing quality traits and estimates of the effects of the important alleles at these loci, can increase the efficiency of genomics assisted breeding in wheat.

Wheat quality, a complex association between many traits and factors, determines the market value of this crop. Although many factors define wheat quality, grain protein content (GPC), grain hardness (GH), flour extraction (FE) or milling yield are foremost determinants. Grain protein content defines the nutritional and end-use properties of dough mixing and rheological characteristics and thus effects the efficiency of the bread making process and product quality [12]. Flour extraction or milling yield is of great economic value. According to some estimates, a 2% reduction in milling yield results in a loss of approximately $7 per ton, which translates to millions of dollars loss every year for the wheat industry [13]. Endosperm texture or grain hardness defines wheat classification and determines the flour extraction and other end use quality parameters [14–16]. At the same time, yield is always a primary target in wheat breeding programs. Grain yield is influenced by many correlated traits including grains per spike (GPS), number of spikes per square meter (SPMS), thousand grain weight (TGW), heading date (HD), and plant height (PH). Because of high economic importance, a number of studies have investigated genetic control of quality [1, 3, 17, 18] and yield related traits [19–24] in wheat. However, majority of those studies were based on low density genetic maps [25], thus limiting their use in marker assisted breeding programs. Most importantly, very limited

or almost no information is available about the genetics of quality and yield traits in hard red spring wheat [25, 26], which is a necessary first step in marker based improvement of those traits in this important wheat class.

Wheat quality and grain yield are affected by grain shape and size. It is believed that large, spherical grains are optimal for milling [27], whereas, small and shriveled grain reduce milling yield (flour extraction and quality). Grain yield is positively affected by grain size as it increases the grain weight [22, 28–30] and promotes seedling vigor [31, 32]. Therefore, obtaining an optimal grain shape and size could be a way to improve grain yield and quality in wheat. However, the genetic association of grain shape and size with wheat quality and yield is very poorly studied. Therefore, in the present study, a recombinant inbred line (RIL) population developed from a cross of an elite HRSW and a non-adapted wheat genotype was used to 1) dissect the genetics of wheat quality and yield traits using a high density iSelect 90K SNP assay based linkage map, 2) identify novel QTL and markers closely associated with quality and yield for marker-assisted breeding, 3) identify the role epistasis in the genetic control of quality and yield and 4) understand the genetic association of wheat quality and yield traits with grain shape and size traits. To the best of our knowledge, this is the first such comprehensive study in hard red spring wheat.

## Material and methods

### Plant material and field evaluation

The present study used an RIL population of 160 lines developed from a cross between an elite line "ND 705" and an exotic line "PI 414566" (or WCB462). The details about the mapping population are provided in Kumar et al. [22]. Briefly, the line ND 705 was developed by the HRSW breeding program at North Dakota State University (NDSU), Fargo, ND, USA, while PI 414566 is originally from China and was obtained from the USDA National Small Grains Collection. PI141566 is a facultative type of tall wheat and has longer grains with high protein content. The RIL population was advanced through single seed descent (SSD) method to $F_8$ generation. The plant material (160 RILs, two parental genotypes and seven checks) was evaluated for two (2009, 2010) years at two locations/year (Prosper and Carrington) of North Dakota, USA. Lines were planted in a $13 \times 13$ partially balanced square lattice design, with two replicates. Each genotype in a replication was planted in a plot consisting of seven rows of 2.44 m length and a distance of 12.7 cm between rows. Sowing rate was 113 kg ha$^{-1}$ in all environments. All the plant material was harvested only after complete maturity and the seeds were uniformly dried (90˚F) for 2–3 days in a drier, before phenotypic analyses.

### Data collection

The phenotypic data was collected for four environments (1 = Carrington 2009, 2 = Carrington 2010, 3 = Prosper 2009, 4 = Prosper 2010). For each genotype in each replication, the phenotypic data was recorded for following eight traits:

1. Grain protein content (%), measured at 12% moisture basis using Infratec 1226 Cold Grain Analyzer following the AACCI standard method 46–30 [33].

2. Grain hardness (0-extra soft to 90-extra hard), was measured on 300 kernels using the Single Kernel Characterization System (SKCS). The hardness was expressed as an index of 0 (extra soft) to 90 (extra hard).

3. Flour extraction (%), measured using 100 or 150 g grain sample tempered to 16.0% moisture. The grain samples were milled using Brabender Quadrumat Junior Mill. The

bran was discarded from the flour. Flour extraction was reported on clean dry wheat basis [34].

4. Grain yield (kg ha$^{-1}$), was measured for each plot (1.86 sq mt) and converted to kilogram/hectare.

5. Number of spikes (spikes m$^{-2}$), determined by counting the number of stems in two one meter long rows per plot and subsequently converted to stems per square meter.

6. Number of grains (grains spike$^{-1}$), measured as an average number of grains in five spikes per genotype.

7. Plant height (cm), measured at maturity from the soil surface to the tip of the spike, excluding the awns.

8. Days to heading (days), determined as the number of days from planting to heading.

9–14. Details about the phenotypic data for six grain shape and size (grain length, grain width, grain area, grain length width ratio, test weight or grain weight by volume and thousand grain weight) traits are reported earlier in Kumar et al. [22]. Briefly, digital image analysis was used to estimate grain length, width, area and length/width ratio. A digital camera was used to manually capture the images of seeds samples placed within a predefined field of view. The images were then processed using Aphelion software (Amerinex Applied Imaging, Amherst, MA). Using image analysis, grain length (GL) (mm), grain width (GW) (mm), and area (GA) (mm$^2$) were calculated from a 10 g samples for each genotype from each replication [35]. Grain length / width ratio (GLWR) was calculated by dividing the grain length mean by the grain width mean for each genotype. Thousand grain weight (TKW) (g) was calculated using the number of grains in a 10 g sample and test weight (TW) (kg m−3) was measured according to the American Association of Cereal Chemist International (AACCI) method 55–10.01 [33].

## Phenotypic data analysis

The analysis of variance (ANOVA) was conducted using the MIXED procedure of the Statistical Analyses System [36]. Within the mixed model, the RIL, their parents and checks were considered fixed effects, while environments and blocks were considered random effects. As the error variances were homogeneous among the four environments, a combined analysis of variance was conducted for all the traits. Mean separation test was performed using an F-protected least significant difference (LSD) value at $P \leq 0.05$ level of significance for each evaluated trait. Pearson correlations, between different traits and between environments of individual traits, were calculated and plotted in R 3.2.2 (https://www.r-project.org/) using cor.matrix and corrplot from the corrplot package. The frequency distribution of different traits was also plotted using R software. The t-test (assuming unequal variances) function in excel 2013, was used to compare the means of two group of lines differing for any particular QTL alleles associated with a phenotypic trait of interest.

## Framework linkage map

In this study, we used a high density framework linkage map developed using Infinium iSelect 90K assay [22]. Briefly, the linkage maps consist of a total of 10,172 markers mapped onto 21 wheat chromosomes with an average of 484.4 markers per chromosome. The total genetic map length covered by 10,172 markers was 4,676.1 cM, with an average distance of 0.46 cM

between any two markers. The 10,172 markers represented a total of 2,591 unique loci, with an average of 123.4 unique loci per chromosome. The average distance between two unique loci was 1.80 cM. The marker order was found to be consistent when compared with earlier published genetic maps [37]. For QTL mapping in this study, a total of 2,591 unique loci were used (S1 Table).

## QTL mapping

In order to identify main-effect QTL, composite interval mapping (CIM) was conducted using QTL Cartographer V2.5_011 [38]. Quantitative trait loci mapping was conducted using the phenotypic data collected for each trait in individual environment as well for the mean data across environments (M). The CIM was conducted using default settings {model 6 (standard model), forward and backward regression, five control markers (cofactors), windows size of 10 cM, and walk speed of 1 cM} in QTL Cartographer. The threshold LOD scores calculated using 1000 permutations were used to declare a significant QTL, however, if any such significant QTL was identified with LOD below the threshold but >2 in other environments, the QTL were also included in the results as supporting information. Only the QTL identified in at least two data sets {out of five (four environments and mean)} or the QTL associated with at least two phenotypic traits were included in this study. The confidence intervals (CI) using ±1 (from the peak) method were estimated using QTL Cartographer. Any two QTL showing overlapping CIs or located within ~10 cM region were considered as one QTL. Inclusive composite interval mapping (ICIM) using QTL IciMapping V 4.1 software [39] was also used to confirm the main effect QTL identified by Cartographer. For ICIM, only those main effect QTL which were detected above threshold LOD score based on 1,000 permutations, were considered. QTL IciMapping V 4.1 was also used to identify QTL×QTL epistatic interactions via a two-dimensional scan. In the IciMapping V 4.1, digenetic epistatic interactions were identified using ICIM-EPI (QICE model) with a fixed LOD scores of 5.0 and walk speeds of 2 cM. In ICIM, a two-stage stepwise regression strategy is applied to identify the most significant markers and marker-pair multiplications in a linear model. First, the best regression model that properly identifies markers and marker pairs explaining additive and epistatic variations, is selected. In the second stage, two-dimensional scanning or interval mapping approach is applied to the adjusted phenotypic values to identify significant QTL in marker intervals and estimate their effects. The phenotypic values are adjusted by using the regression model selected in the first step. The phenotypic values for interval mapping are adjusted to control genetic background effect, just like using co-factors in CIM by QTL Cartographer. It means that the adjusted values retain the information of QTL on the current mapping interval but exclude the influence of QTL on other intervals and chromosomes [40]. The ICIM approach identifies the epistatic QTL, with or without any significant main effect.

To suggest, if the QTL identified in this study are located in the already reported genomic regions or if they are putative novel QTL, we compared the genomic locations of the identified QTL with earlier studies using associated markers. It may be noted that the genetic map used in this study was based on only SNP markers. However, many earlier studies are based on SSRs or DArT markers. Therefore, the genetic maps which contain both SSRs and SNPs from Infinium 9K or 90K assays [29, 30, 41] or SSRs and DArT markers were used as a bridge to compare the locations of QTL identified in this study with earlier reported studies. If needed, the information about SSR markers (closely mapped to the QTL associated SSRs) was also obtained from GrainGenes website (http://wheat.pw.usda.gov/cgi-bin/graingenes/browse.cgi?class=marker).

## Results

### Phenotypic variation

The RIL population showed continuous variation for all the eight traits, suggesting a polygenic inheritance (Fig 1). Among the parents, the mean data showed that elite genotype ND 705 had higher values for FE, GPC, GH, GY, SPMS, GPS, PH, and lower values for DH (Table 1; Fig 1). The same trend was observed in individual environments for all the traits with few exceptions (S2 Table). Transgressive segregation in the desirable direction was observed in almost all the environments for all the traits in the RIL population (Fig 1) suggesting the presence of superior QTL alleles in both the parents. The ANOVA showed significant differences among the genotypes as well as significant genotype × environment interaction for all the traits. In general, all the traits showed higher values in Carrington compared to Prosper (S2 Table).

### Phenotypic correlation

Phenotypic association between quality, yield and grain shape and size traits were also investigated (Fig 2). Among the three quality traits, FE and GH showed positive and significant (P = 0.05) correlation (0.24 to 0.49). However, FE, GPC showed a weak negative correlation (-0.13 to -0.33). Grain protein content and GH showed both positive and negative correlations in different environments. The correlations between grain quality traits (FE, GPC and GH) and GY were mostly non-significant. Grain yield had non-significant correlation with GPC, except for one location (Prosper 2010), where it showed a weak negative correlation (-0.29). Grain yield was positively correlated with yield components number of spikes per square meter (SPMS) and GPS (grains per spike) in all the environments, although the values were small. Days to heading and PH showed both positive and negative correlation with GY, although the association was weak.

Grain shape and size traits (GL, GW, GLWR, GA, TGW and TW) showed both positive and negative correlations with quality and agronomic traits. However, the correlations were mostly weak. Grain shape and size traits had positive but low correlation with flour extraction, low to moderate negative correlations with grain hardness, and both positive and negative correlation with grain protein (depending on location). Grain width, GA, TGW and TW showed positive low to moderate correlation with GY, depending upon the year and location. Test weight had a significant positive effect on GY in all the environments. The associations were weak in 2009, but moderate in 2010. Spikes per square meter, grain number per spike and thousand grain weight are considered three major components of yield. In this study, although these three yield components showed positive correlation with yield, only spikes per square meter, grain number per spike showed significant correlation with yield in all the environments (Fig 2). Thousand grain weight (TGW) showed significant correlation at both location in only in the year 2010, suggesting that among the three important components of yield in wheat, TGW is probably most influenced by the environment conditions. Among the top five high yielding RILs (30, 86, 88, 92 and 143), which showed higher yield than both the parents, all five had more grains per spike; three lines had more spikes per square meter and two lines had higher TGW, when compared to the parental genotypes. This suggests the importance of these component traits in enhancing yield in wheat.

### QTL analysis

Composite interval mapping for eight traits dissected in this study identified a total of 60 main effect QTL located in 43 genomic regions belonging to 19 different chromosomes (all except 1D and 4A) (Table 2; S3 Table). The majority (52 out of 60) of these QTL were also identified

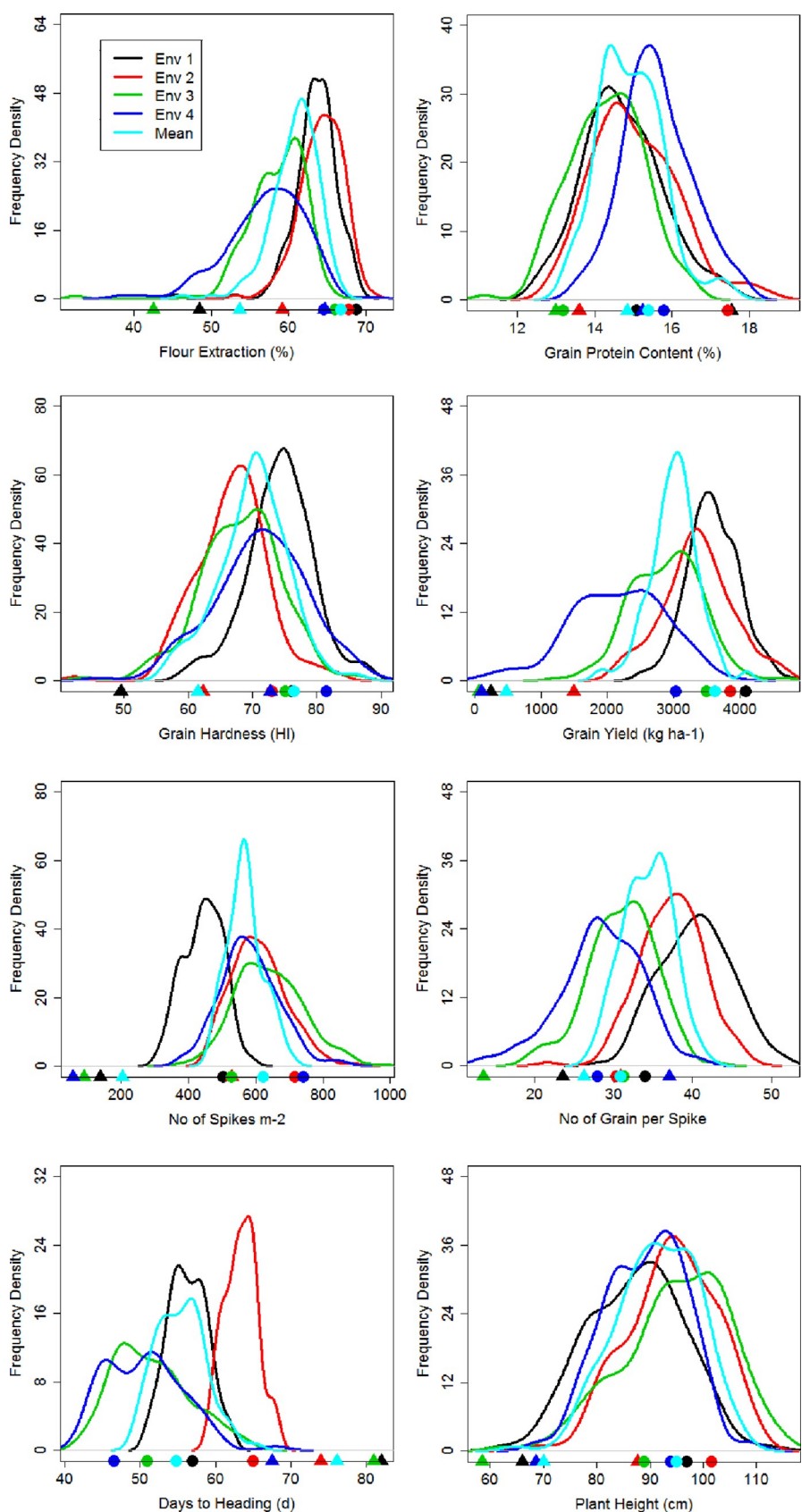

**Fig 1. Frequency distribution of 160 RILs of ND 705 (elite genotype) and PI 414566 (non-adapted genotype) population for mean of eight quality, yield and agronomic traits evaluated over four environments.** The mean of the parental genotypes in different environments is indicated by a circle (ND 705) and a triangle (PI 414566).

by IciMapping. The few QTL which were only detected by CIM (using QTL cartographer) were specific to individual environments (Table 2).

## Quality traits {flour extraction (FE), grain protein content (GPC), grain hardness (GH)}

Composite interval mapping (CIM) identified 11, 14 and seven QTL for FE, GPC and GH respectively, in this population (Table 2). The 32 QTL identified for three quality traits were located in 28 genomic regions (S3 Table). Four genomic regions (3, 10, 39, and 42) harbored QTL for two traits. The phenotypic variation explained (PVE) by individual QTL for quality traits ranged from 5.6 to 24.9%.

A total of four, eight and six QTL for FE, GPC and GH respectively, were identified in at least three (out of five) datasets and could be considered stable QTL. The most important QTL identified for FE was located on 5BL (*QFE.ndsu.5B*) in a 2.7 cM interval (280.6–283.3 cM on 5B). This QTL had both stable and major effect (PVE upto 12.6%) on FE and ND 705 alleles increased the trait value. Interestingly, PI 414566 also contributed alleles for increased FE at two stable loci (located on 3A and 5A) (Table 2). For GPC, the two most stable and major QTL were identified one each on 7A and 7B; both being independent of yield. The QTL located on the long arm of 7A (*QGPC.ndsu.7A.2*) was consistently detected in all the environments as well in the mean data, and contributed up to 14.6% of PV for GPC. The QTL located on short arm of 7B (*QGPC.ndsu.7B*) had the most significant effect on GPC and was also detected in four of the five data sets. Most importantly, the positive alleles for increased grain protein for six (out of 8) stable QTL, including the important loci on 7A and 7B, were contributed by PI 414566, the non-adapted genotype. For GH, one QTL located on 4B (*QGH.ndsu.4B.1*) was detected above threshold in all the five data sets and contributed up to 10.5% of PV for GH. The elite genotype ND 705 contributed alleles for increased hardness at this locus. Other two most consistent QTL with major effect on GH were detected one each on 1A (*QGH.ndsu.1A*) and 1B (*QGH.ndsu.1B*), but the alleles for increased hardness at those two loci were contributed by PI 414566.

## Yield traits {grain yield (GY), grains per spike (GPS) and spikes per square meter (SPMS)}

Three yield traits, grain yield (GY), grains per spike (GPS) and spikes per square meter (SPMS) were included in this study. Composite interval mapping identified a total of five QTL

**Table 1. Mean phenotypic performance of elite genotype ND705, non-adapted genotype PI 414566, and their recombinant inbred lines (RILs) across four environments.**

| Trait | RILs | | | Parental genotypes | | LSD$_{0.05}$ | CV (%) |
|---|---|---|---|---|---|---|---|
| | Min | Max | Mean | ND705 | PI414566 | | |
| Flour extraction (%) (FE) | 46.3 | 66.7 | 60.8 | 66.8 | 53.7 | 5.4 | 3.3 |
| Grain protein content (%) (GPC) | 13.3 | 17.6 | 14.9 | 15.4 | 14.9 | 0.7 | 3.5 |
| Grain hardness (HI) (GH) | 56.7 | 86.6 | 70.1 | 76.4 | 61.5 | 4.1 | 3.1 |
| Grain yield (kg ha-1) (GY) | 1843.7 | 4119.1 | 3001.4 | 3624.6 | 474.5 | 610 | 11.7 |
| No of spikes m$^2$ (SPMS) | 452.1 | 717.7 | 569.0 | 623.2 | 203.8 | 190 | 15.9 |
| No of Grains per spike (GPS) | 27.0 | 42.2 | 34.3 | 30.9 | 26.2 | 7.2 | 10.6 |
| Days to heading (d) (DH) | 49.4 | 65.5 | 55.4 | 54.9 | 76.1 | 2.1 | 1.9 |
| Plant height (cm) (PH) | 64.8 | 107.6 | 91.4 | 95.3 | 70.2 | 7.6 | 4.3 |

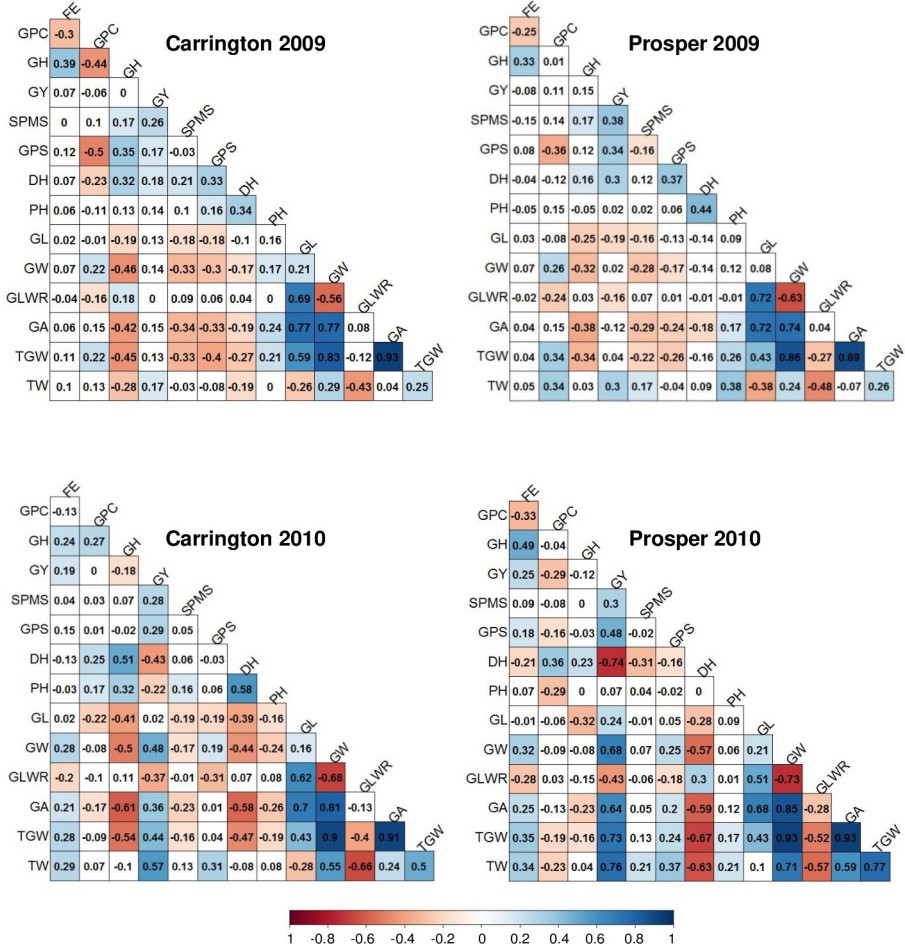

**Fig 2. Phenotypic correlations for quality, yield, agronomic and grain shape and size traits in ND 705 (elite genotype) × PI 414566 (non-adapted genotype) RIL population.** FE = Flour extraction (%), GPC = Grain protein content (%), GH = Grain hardness (HI), GY = Grain yield (kg ha-1), SPMS = No of spikes m2, GPS = No of Grains per spike, DH = Days to heading (d), PH = Plant height (cm), GH = Grain length, GW = Grain width, GA = Grain area, GLWR = Grain length width ratio, TGW = Thousand grain weight, TW = Test weight.

each for GY and GPS and seven QTL for SPMS in this population (Table 2; S3 Table). The 17 QTL were located in 14 genomic regions; one genomic region (genomic region 19) had QTL for two and another (genomic region 17) had QTL for three traits (S3 Table). Overall, the PVE by these QTL ranged from 7.5 to 19.7%. Four QTL for GY, three QTL for GPS and five QTL for SPMS explained >10% PV and could be considered major. For GY, one QTL located on 4B (*QGY.ndsu.4B*) was detected in three data set and could be considered stable; all other QTL were detected in one or two data sets. The elite genotype ND 705 contributed the alleles for increased GY at all the identified loci (Table 2). For GPS, the QTL located on 2B (*QGPS. ndsu.2B*) and 3D (*QGPS.ndsu.3D*) were detected in four environments and could be considered stable. Elite line ND 705 contributed the alleles for increased GPS at both *QGPS.ndsu.2B* and *QGPS.ndsu.3D*. The RILs with ND 705 alleles at both *QGPS.ndsu.2B* and 3D *QGPS. ndsu.3D* had an average 36.1 grains per spike, while the lines with the PI 414566 alleles at both those loci showed an average of 33.1 grains per spike. The RILs with the ND 705 alleles at *QGPS.ndsu.2B* locus and the PI 414566 allele at *QGPS.ndsu.3D* had an average 34.7 grains per spike, whereas, lines with the PI 414566 allele at *QGPS.ndsu.2B* locus and the ND 705 allele at

**Table 2. Quantitative trait loci identified for grain quality and yield related traits and their association with grain shape and size related traits in a hexaploid wheat RIL population derived from the cross of an elite genotype (ND 705) and a non-adapted genotype (PI 414566).**

| QTL Region | QTL name | Other Associated traits | Env. | Position(cM) | LOD | Add. effect | R2 (%) |
|---|---|---|---|---|---|---|---|
| **Flour extraction (FE)** | | | | | | | |
| 3 | QFE.ndsu.1A | GPC | 2 | 167.2 | 4.9 | 1.05 | 10.7 |
| 10 | QFE.ndsu.2A | GPC | 4 | 108.2 | 4.1 | 1.47 | 7.7 |
| 14 | QFE.ndsu.3A2.1 | SPMS | 1,4,5 | 68.1–75.5 | 5.4 | -1.46 | 9.9 |
| 15 | QFE.ndsu.3A2.2 | | 1,5* | 120.6–121.6 | 6.3 | 1.04 | 11.4 |
| 16 | QFE.ndsu.3B | | 1*,2*,5 | 124.4–142.0 | 3.5 | 0.80 | 6.3 |
| 21 | QFE.ndsu.4D | | 3,5 | 0.0–2.0 | 4.1 | 1.00 | 6.9 |
| 25 | QFE.ndsu.5A.1 | | 4,5 | 151.4 | 5.2 | 1.67 | 9.8 |
| 29 | QFE.ndsu.5A.2 | | 1,2,5* | 343.5–349.8 | 4.4 | -0.99 | 9.4 |
| 32 | QFE.ndsu.5B | | 1,2,3,5* | 280.6–283.3 | 5.9 | 1.40 | 12.6 |
| 33 | QFE.ndsu.5D2 | | 3*,4 | 4.2–6.7 | 4.1 | 1.41 | 7.6 |
| 42 | QFE.ndsu.7D | GPC | 3 | 2.7 | 5.2 | 1.07 | 7.3 |
| **Grain protein content (GPC)** | | | | | | | |
| 3 | QGPC.ndsu.1A | FE | 1*,4,5* | 175.1–188.1 | 5.4 | -0.30 | 10.5 |
| 5 | QGPC.ndsu.1B.1 | GLWR | 1*,4*,5 | 93.7 | 3.9 | 0.22 | 5.6 |
| 8 | QGPC.ndsu.1B.2 | | 2*,4,5 | 210.3–223.4 | 5.1 | -0.29 | 9.3 |
| 10 | QGPC.ndsu.2A$ | FE | 3 | 114.5 | 4.1 | 0.27 | 7.5 |
| 19 | QGPC.ndsu.4B$ | GY,SPMS, GL, GA, TW, TGW | 1 | 83.0 | 3.7 | 0.25 | 5.6 |
| 22 | QGPC.ndsu.4D | GPS | 1*,2,5 | 45.3–48.4 | 9.2 | 0.45 | 15.1 |
| 27 | QGPC.ndsu.5A.1 | DH,SPMS | 4 | 218.9 | 4.6 | -0.34 | 12.9 |
| 28 | QGPC.ndsu.5A.2 | | 2,3*,5 | 248.9–269.3 | 6.1 | -0.38 | 10.7 |
| 30 | QGPC.ndsu.5B | | 1,2*,3*,5 | 25.6–27.6 | 8.5 | -0.44 | 17.5 |
| 36 | QGPC.ndsu.6D2 | SPMS | 2*,3 | 30.5–40.1 | 5.2 | -0.35 | 11.6 |
| 38 | QGPC.ndsu.7A.1$ | | 1,5* | 110.3–110.9 | 8.8 | 0.41 | 14.6 |
| 39 | QGPC.ndsu.7A.2 | GH | 1*,2,3,4,5 | 363.8–369.1 | 8.8 | -0.45 | 14.6 |
| 40 | QGPC.ndsu.7B | | 1,2,4*,5 | 7.7 | 8.6 | -0.5 | 24.9 |
| 42 | QGPC.ndsu.7D1 | FE | 2 | 7.9 | 4.1 | 0.30 | 6.5 |
| **Grain hardness (GH)** | | | | | | | |
| 1[E] | QGH.ndsu.1A | | 2,3,4,5 | 5.3–7.9 | 11.4 | -2.58 | 21.3 |
| 6 | QGH.ndsu.1B | PH, GW, GLWR | 1,2,3,5 | 137.1–155.9 | 6.4 | -1.81 | 11.8 |
| 18 [E] | QGH.ndsu.4B.1 | | 1,2,3,4,5 | 38.2–49.1 | 6.0 | 2.26 | 10.5 |
| 20 | QGH.ndsu.4B.2 | | 2,4*,5 | 183.4 | 5.1 | 1.70 | 8.5 |
| 24 | QGH.ndsu.5A.1 | | 4,5* | 110.9 | 4.3 | 2.09 | 7.5 |
| 26 | QGH.ndsu.5A.2 | | 1,2,5* | 185.8–187.7 | 6.6 | 1.89 | 12.9 |
| 39 | QGH.ndsu.7A | GPC | 1,2*,4*,5 | 364.8–373.1 | 4.2 | 1.50 | 7.4 |
| **Grain yield (GY)** | | | | | | | |
| 7 | QGY.ndsu.1B | DH,PH | 3*,4 | 183.5, 203.3 | 5.1 | 207.94 | 10.2 |
| 17 | QGY.ndsu.3D1 | GPS, SPMS, PH, TGW, GA | 2 | 8.0 | 6.3 | 187.36 | 11.1 |
| 19 | QGY.ndsu.4B | GPC,SPMS, GL, GA, TW, TGW | 2,3*,5* | 88.5 | 6.5 | 185.18 | 11.4 |
| 34 | QGY.ndsu.6A | GA, TGW | 3,5 | 141.5–146.0 | 6.9 | 199.31 | 13.3 |
| 35 | QGY.ndsu.6B | GL | 3,5 | 90.3–92.3 | 4.2 | 151.52 | 7.5 |
| **Grains per spike (GPS)** | | | | | | | |
| 4 | QGPS.ndsu.1B$ | | 1*,3 | 25.7–31.5 | 4.3 | 1.31 | 9.1 |
| 9 | QGPS.ndsu.2A | PH, GA | 3,5 | 14.4–16.9 | 5.7 | -1.53 | 12.2 |
| 11 | QGPS.ndsu.2B | | 1*,2,4,5* | 190.1–199.3 | 5.1 | 1.71 | 10.4 |
| 17 | QGPS.ndsu.3D1 | GY,SPMS,PH, TGW, GA | 1*,2,3*,5* | 12.8–32.3 | 4.4 | 1.17 | 8.1 |

*(Continued)*

**Table 2.** (Continued)

| QTL Region | QTL name | Other Associated traits | Env. | Position(cM) | LOD | Add. effect | R2 (%) |
|---|---|---|---|---|---|---|---|
| 22 | QGPS.ndsu.4D | GPC | 2 | 47.0 | 5.5 | -1.33 | 10.2 |
| **No of spikes per sq mt (SPMS)** | | | | | | | |
| 2 | QSPMS.ndsu.1A | | 2,5 | 94.8 | 9.6 | 38.68 | 19.7 |
| 14 | QSPMS.ndsu.3A$ | FE | 3 | 75.5 | 4.1 | 30.95 | 9.1 |
| 17 | QSPMS.ndsu.3D1 | GY,GPS,PH, TGW, GA | 2,5* | 17.6,30.3 | 5.4 | 26.77 | 10.5 |
| 19 | QSPMS.ndsu.4B | GPC,GY, GL, TW, TGW, GA | 2,5 | 80.1–81.1 | 5.1 | 18.52 | 10.0 |
| 23 | QSPMS.ndsu.4D | | 4,5* | 58.4 | 7.2 | 37.33 | 16.2 |
| 27 | QSPMS.ndsu.5A$ | GPC,DH, GA, TGW | 4 | 217.9 | 5.0 | 37.67 | 16.0 |
| 36 | QSPMS.ndsu.6D$ | GPC | 3 | 25.1 | 3.6 | -29.81 | 7.9 |
| **Heading date (DH)** | | | | | | | |
| 7 | QDH.ndsu.1B | GY,PH | 4,5 | 179.5–184.5 | 8.7 | -1.84 | 12.6 |
| 27 | QDH.ndsu.5A | GPC,SPMS | 1,2*,3,4,5 | 220.9–224.9 | 18.6 | -3.64 | 38.8 |
| 31 | QDH.ndsu.5B | | 1*,2,3*,4,5 | 132.6–138.8 | 7.3 | -1.77 | 10.8 |
| 43 | QDH.ndsu.7D2 | | 2,3,5 | 16.9–17.9 | 9.0 | 2.06 | 15.6 |
| **Plant height (PH)** | | | | | | | |
| 7 | QPH.ndsu.1B | GY,DH | 1*,2*,3 | 185.1–194.4 | 6.9 | -3.55 | 12.1 |
| 9 | QPH.ndsu.2A | GPS | 4 | 5.0 | 4.8 | -2.20 | 7.6 |
| 12 | QPH.ndsu.2D1 | | 1,2,3,4,5 | 34.9 | 17.1 | 5.20 | 31.2 |
| 13 | QPH.ndsu.3A$ | | 1*,3*,5 | 29.3–32.6 | 6.2 | 2.38 | 9.0 |
| 17 | QPH.ndsu.3D1 | GY,GPS,SPMS | 1,2,4*,5 | 39.9 | 5.6 | -2.63 | 9.1 |
| 37 | QPH.ndsu.7A | | 1,2* | 15.7–20.8 | 4.1 | -2.14 | 5.3 |
| 41 | QPH.ndsu.7B | | 1,2,5* | 121.3–123.7 | 6.0 | 2.62 | 9.5 |

[E] QTL also involved in QQ epistatic interaction

$QTL only detected by QTL Cartographer; all other QTL were detected by both QTL Cartographer and QTL ICI mapping

FE = Flour extraction (%), GPC = Grain protein content (%), GH = Grain hardness (HI), GY = Grain yield (kg ha-1), SPMS = No of spikes m2, GPS = No of Grains per spike, DH = Days to heading (d), PH = Plant height (cm), GH = Grain length, GW = Grain width, GA = Grain area, GLWR = Grain length width ratio, TGW = Thousand grain weight, TW = Test weight

*QTL detected above 2 LOD but below threshold LOD estimated by permutations

*QGPS.ndsu.3D* had an average of 34.1 grains per spike. This means, the ND 705 alleles at both *QGPS.ndsu.2B* and *QGPS.ndsu.3D* provide an advantage of three grains per spike. None of the QTL for SPMS, could be detected in two environments, suggesting that they all are unstable QTL.

## Agronomic traits {days to heading (DH), plant height (PH)}

This study identified four QTL for DH and seven QTL for PH (Table 2; S3 Table). Individually, these QTL contributed 10.8–38.8% and 5.3 to 31.2% of the phenotypic variation for DH and PH, respectively. For DH, the QTL identified on 5A (*QDH.ndsu.5A*) could be considered most important in this population, as it was detected in all the environments and explained up to 38.8% of PV. Another stable QTL identified in all the environment was located on 5B (*QDH.ndsu.5B*) and contributed up to 10.8% PV for DH. The QTL located on 7D has also major effect (PVE up to 15.6%) on DH and could also be considered stable as it was detected in three data sets. The elite line ND 705 contributed alleles for fewer days to heading at three loci (out of four), including the stable loci located on 5A and 5B (Table 2).

For PH, the most significant QTL (*QPH.ndsu.2D*) in this population was identified on chromosome arm 2DS. The QTL was located in a 3.5 cM interval flanked by BS00067046_51 from

one side and six Co-localized markers on the other side (*Kukri_c51992_290*, *Kukri_rep_ c106786_230*, *wsnp_Ex_c14779_22892053*, *wsnp_JD_rep_c63957_40798083*, *wsnp_JD_rep_ c63957_40798121*, *Kukri_rep_c113120_104*). These six co-localized markers were mapped at a distance of 0.5 cM from the peak. This QTL was detected above threshold LOD score with a consistent phenotypic variation (26.1–31.2%) in all the environments. Interestingly, the alleles for reduced height at this locus were contributed by the non-adapted parent PI 414566 and explained up to 31.2% phenotypic variation for PH. When the whole population was divided into two groups based on parental alleles at *QPH.ndsu.2D* locus, PI 414566 alleles reduced plant height by 8–9.4 cm in different environments with an average of 8.9 cm (Table 3). Four other QTL located one each on 1B, 3A, 3D and 7B were identified for PH in three or four data sets and could be considered stable. Overall, ND 705 contributed reduced height alleles at four loci, including two stable loci. However, at both of these loci, the alleles for reduction in height were associated with reduced yield. The non-adapted parent PI 414566 contributed reduced height alleles at three loci; all of them being stable across environments and independent of yield QTL.

**Table 3. Parental alleles associated with increased/decreased trait values and significant mean differences observed in the RILs[1] for phenotypic trait values when a particular parental allele is present at major QTL identified in this study.**

| Trait/QTL | [2]Allele for increased trait value | Trait value | Allele for decreased trait value | Trait value | t-test for mean difference (P-value) |
|---|---|---|---|---|---|
| **Flour Extraction (%)** | | | | | |
| QFE.ndsu.5B | AA | 58.1 | BB | 55.8 | 0.002556 |
| QFE.ndsu.3A.1 + QFE.ndsu.5A.2 | BB+BB | 62.6 | AA+AA | 59.2 | 0.000001 |
| QFE.ndsu.5B+ QFE.ndsu.3A.1 + QFE.ndsu.5A.2 | AA+BB+BB | 63.0 | BB+AA+AA | 58.9 | 0.000000 |
| **Grain protein content (%)** | | | | | |
| QGPC.ndsu.5B | BB | 15.2 | AA | 14.7 | 0.000493 |
| QGPC.ndsu.7A.2 | BB | 15.3 | AA | 14.7 | 0.000009 |
| QGPC.ndsu.7B | BB | 15.4 | AA | 14.7 | 0.000000 |
| QGPC.ndsu.7A.2 + QGPC.ndsu.7B | BB+BB | 15.9 | AA+AA | 14.5 | 0.000000 |
| QGPC.ndsu.5B + QGPC.ndsu.7A.2 + QGPC.ndsu.7B | BB+BB+BB | 16.2 | AA+AA+AA | 14.5 | 0.000001 |
| **Grain Hardness** | | | | | |
| QGH.ndsu.1A | BB | 72.2 | AA | 68.1 | 0.000000 |
| QGH.ndsu.1B | BB | 71.9 | AA | 68.6 | 0.000180 |
| QGH.ndsu.4B.1 | AA | 70.5 | BB | 69.1 | 0.054342 |
| QGH.ndsu.1A+QGH.ndsu.1B +QGH.ndsu.4B.1 | BB-BB-AA | 75.0 | AA-AA-BB | 60.7 | 0.000000 |
| **Grains per spike** | | | | | |
| QGPS.ndsu.2B | AA | 35.4 | BB | 33.5 | 0.000045 |
| QGPS.ndsu.3D | AA | 35.2 | BB | 33.8 | 0.002273 |
| QGPS.ndsu.2B + QGPS.ndsu.3D | AA+AA | 36.1 | BB+BB | 33.1 | 0.000007 |
| **Heading date (no of days)** | | | | | |
| QDH.ndsu.5A | BB | 57.9 | AA | 54.2 | 0.000000 |
| QDH.ndsu.5B | BB | 55.8 | AA | 55.1 | 0.069471 |
| QDH.ndsu.5A+ QDH.ndsu.5B | BB+BB | 56.7 | AA+AA | 52.5 | 0.000006 |
| **Plant Height (cm)** | | | | | |
| QPH.ndsu.2D | AA | 95.2 | BB | 86.3 | 0.000000 |

[1] Only the RILs showing clear genotype for all the associated markers at the target QTL were used for the analysis

[2]AA = ND 705 allele; BB = PI 414566 allele

## Genetic relationship between grain quality, yield and agronomic traits

Among 43 genomic regions associated with quality, yield and agronomic traits in this population, only five genomic regions (14, 19, 22, 27, and 36; Table 2, S3 Table) harbored QTL for quality related trait(s) as well QTL for yield and/or agronomic trait (s). Majority (80%) of the QTL were either associated with quality or yield or agronomic traits which suggests an independent genetic control for those traits. Among the five regions (14, 19, 22, 27, 36) harboring QTL for quality as well as yield (and agronomic) traits, two regions (19, 36) have desirable alleles for quality and yield traits from the same parent, while in three genomic regions (14, 22, 27) the desirable alleles for quality and yield trait (s) were contributed by different parents. The genomic region 14 (3A) harbored a stable QTL for FE with PI 414566 contributing alleles for higher flour extraction. However, the same region was associated with an unstable QTL for SPMS, with PI 414566 contributing undesirable alleles. The genomic regions 19 (4B) had QTL for GY, SPMS and GPC and the desirable alleles for all three traits were contributed by ND 705. The genomic region 22 (4D) had a stable QTL for GPC with ND 705 increasing the trait values. However, the same region was associated with an unstable QTL for GPS with ND 705 allele responsible for reduced GPS. The genomic region 27 (5A) was associated with a QTL for DH identified in all the environments and a QTL each for SPMS and GPC, identified in only single environment. The ND 705 alleles at this region reduced DH, increased SPMS but were also associated with reduced GPC. The genomic region 36 was associated with an environment specific QTL for SPMS and GPC, with PI 414566 alleles increasing the values for both the traits.

## Genetic association of grain quality, and yield traits with grain shape and size traits

Out of 28 genomic regions associated with three wheat quality traits (GPC, GH, FE) in this study, only five genomic regions (17.8%; S3 Table) were associated with grain shape and size traits. These genomic regions were 5 (1B), 6 (1B), 19 (4B), 27 (5A) and 39 (7A). Among the quality traits, four GPC QTL and two GH QTL (one being common for GPC and GH), were co-localized with QTL for grain shape and size. None of the QTL identified for FE showed any association with grain shape and size traits. On the other hand, co-localization of QTL for yield related traits with grain shape and size was more frequent (Table 2; S3 Table). Out of the 14 genomic regions associated with three yield traits (GY, GPS, SPMS), six (~ 43%) genomic regions were also associated with grain shape and size traits in the same population. These genomic regions were 9 (2A), 17 (3D), 19 (4B), 27 (5A), 34 (6A), and 35 (6B).

The loci for grain quality and yield traits showed both positive and negative associations with grain shape and size traits. The ND 705 allele at the genomic region 27 (5A) was associated with 3.64 days reduction in heading as well as significant increase in GA and TGW. The same parental allele was also responsible for increase in SPMS and decrease in GPC, but SPMS and GPC QTL were significant in only one environment. The genomic region 39 (7A) seems to be a another good candidate for breeding as the PI 414566 allele at this locus was associated with increase in GPC as well as TGW, GL, GW, GA. The same alleles were, however associated with minor effect on grain softness. All the QTL in this region on 7A were stable across environments. In the genomic region 23 (6A), GA was positively associated with GY and TGW, with ND 705 providing the desirable alleles.

## QTL×QTL epistatic interactions

Inclusive composite interval mapping identified a total of 359 digenic epistatic interaction for 14 different traits (including six grain shape and size traits) at a minimum LOD score of five

**Table 4. Summary of main-effect and digenic epistatic interactions detected by inclusive composite interval mapping in a hexaploid wheat RIL population derived from the cross of an elite genotype (ND 705) and a non-adapted genotype (PI 414566).**

| Trait/ Environment | Total no. | | | | | Total $R^2$ (%) | | | | |
|---|---|---|---|---|---|---|---|---|---|---|
| | Carr09 | Carr10 | Pros09 | Pros10 | Mean | Carr09 | Carr10 | Pros09 | Pros10 | Mean |
| **Main effect QTL** | | | | | | | | | | |
| FE | 7 | 3 | 8 | 4 | 8 | 52.5 | 25.4 | 43.1 | 38.2 | 57.1 |
| GPC | 10 | 8 | 5 | 4 | 9 | 69.0 | 62.5 | 43.0 | 38.8 | 70.5 |
| GH | 7 | 5 | 4 | 2 | 7 | 52.8 | 47.5 | 40.8 | 26.8 | 56.0 |
| YLD | 1 | 6 | 2 | 2 | 3 | 8.4 | 51.2 | 25.6 | 20.1 | 39.2 |
| GPS | 2 | 7 | 1 | 2 | 2 | 23.1 | 56.0 | 14.7 | 23.5 | 26.0 |
| SPMS | - | 6 | - | 2 | 4 | - | 44.1 | - | 16.4 | 34.8 |
| DH | 2 | 2 | 2 | 9 | 4 | 27.7 | 11.4 | 37.9 | 73.3 | 54.5 |
| PH | 5 | 2 | 3 | 3 | 8 | 57.2 | 32.7 | 39.8 | 42.8 | 66.8 |
| KL | 9 | 7 | 10 | 4 | 18 | 63.5 | 55.0 | 72.3 | 41.9 | 86.5 |
| KW | 7 | 6 | 4 | 4 | 9 | 61.2 | 57.3 | 42.6 | 33.8 | 71.2 |
| KA | 10 | 8 | 7 | 6 | 9 | 68.7 | 57.2 | 57.3 | 49.3 | 66.3 |
| KLWR | 10 | 10 | 10 | 11 | 17 | 72.9 | 73.6 | 68.5 | 77.7 | 88.8 |
| TKW | 7 | 8 | 5 | 4 | 9 | 49.4 | 56.5 | 46.5 | 39.1 | 61.0 |
| TW | 5 | 4 | 3 | 5 | 5 | 42.7 | 45.1 | 31.5 | 41.6 | 49.5 |
| **Epistatic interactions** | | | | | | | | | | |
| FE | 5 | 29 | 3 | 10 | 6 | 59.5 | 68.6 | 28.2 | 28.6 | 18.8 |
| GPC | 6 | 10 | 3 | 8 | 5 | 60.8 | 58.8 | 25.3 | 49.6 | 56.9 |
| GH | 17 | 6 | 21 | 16 | 8 | 66.7 | 56.3 | 74.3 | 63.7 | 56.4 |
| YLD | 2 | 6 | 4 | 7 | 7 | 25.2 | 57.8 | 42.0 | 45.2 | 56.6 |
| GPS | 4 | 4 | - | 1 | 2 | 54.5 | 35.4 | - | 11.0 | 27.2 |
| SPMS | 9 | - | - | - | - | 49.1 | - | - | - | - |
| DH | 12 | 11 | 10 | 8 | 5 | 63.4 | 45.3 | 57.2 | 53.2 | 60.1 |
| PH | - | 3 | 2 | 10 | 7 | - | 27.4 | 20.5 | 20.4 | 25.6 |
| KL | 4 | 4 | 1 | 6 | 3 | 37.8 | 35.2 | 11.1 | 49.9 | 40.7 |
| KW | 3 | 4 | 4 | 4 | - | 36.1 | 43.6 | 37.9 | 49.0 | - |
| KA | 1 | - | - | - | - | 10.5 | - | - | - | - |
| KLWR | 8 | - | 5 | 4 | 1 | 36.3 | - | 52.7 | 36.6 | 7.3 |
| TKW | 1 | 3 | 5 | 5 | - | 13.9 | 45.7 | 45.8 | 44.8 | - |
| TW | 8 | 4 | 10 | 8 | 5 | 47.7 | 34.8 | 51.6 | 33.3 | 28.1 |

FE = Flour extraction (%), GPC = Grain protein content (%), GH = Grain hardness (HI), GY = Grain yield (kg ha-1), SPMS = No of spikes m2, GPS = No of Grains per spike, DH = Days to heading (d), PH = Plant height (cm), GH = Grain length, GW = Grain width, GA = Grain area, GLWR = Grain length width ratio, TGW = Thousand grain weight, TW = Test weight

(Table 4; S4 Table). The total number of epistatic interactions for an individual trait in a single environment ranged from zero to 29. For SPMS and KA, epistasis was detected only in one dataset (out of 5), while for the rest of the traits, epistasis was detected in at least four datasets. More epistatic interactions were identified for quality traits (FE, GPC and GH) compared to yield or grain shape and size related traits. The phenotypic variation (PV) explained by all the epistatic interaction belonging to a single trait in individual environments ranged from 7.3 to 74.3%. The average PV explained by all epistatic interactions for a trait in a single environment was 41.8%, compared to 48.2% for the main effect QTL, suggesting the importance of epistasis in the genetic control of different traits.

Although a large number of the epistatic interactions were associated with individual traits, a total of 51 epistatic interactions representing 20 unique QQ pairs were found associated with

two or more environments (called stable QQ interactions) and/or traits (Table 5). The maximum number of six stable QQ interactions were detected for grain hardness, followed by four stable interactions for DH. For FE, KL, KW and TW, one QTL interaction was detected for each trait in two or more environments (Table 5). Both parental (two loci from the same parent increasing the trait values) and recombinant (one locus from each parent increasing the trait values) QQ interactions were observed. The QTL×QTL digenic interaction could involve two main-effect QTL (M-QTL), an M-QTL and an epistatic QTL (E-QTL; a QTL with no main effect), or two E-QTL. In the present study, almost all the QQ interactions were observed between E-QTL. Such QTL will escape detection in the analyses, which detect interactions between M-QTL only.

## Discussion

Hard red spring wheat is considered a specialty wheat because of its high protein content and quality, traits which help produce excellent quality bread and determine the marker value of the crop. Therefore, grain protein content and flour extraction are the major targets of wheat breeding programs. At the same time, improvement in grain yield is always the primary target in any breeding program. This study investigates the high resolution genetic dissection of quality, yield and agronomic traits in hard red spring wheat to identify new useful alleles from adapted and non-adapted genotypes. The study also provides insight into the genetic association of grain quality and yield traits with grain shape and size traits. The population used is this study was derived from an elite line ND 705 and a non-adapted line PI 414566, and thus segregates for yield, quality and grain shape and size traits. The non-adapted germplasm is required to identify new sources of desirable alleles to enrich the cultivated gene pool. The cross, therefore, resulted in the identification of desirable alleles fixed in the breeding program as well as novel and desired alleles contributed by the un-adapted genotype. The use of high density Infinium 90k assay based genetic map allowed the identification of tightly linked markers with the traits of interest, which is critical for the success of marker assisted selection.

### Novel QTL for flour extraction (FE)

The few studies conducted in the past to genetically dissect FE were based on low resolution genetic maps [17, 25, 42, 43]. Although multiple QTL were associated with FE in our study, the inconsistency of most of them across environments suggests a significant environmental effect, as has been reported in the past [17]. The QTL *QFE.ndsu.5B* seems to be the most significant for FE as it was detected across multiple environments. The separation of RILs based on parental alleles at this locus showed that the ND 705 allele increased FE by 2% (61.8 *vs* 59.8%). The QTL for milling or flour yield were also reported on 5B in earlier studies [17, 25, 42]. However, when compared with those studies, the location of these QTL seems far from *QFE.ndsu.5B*, suggesting it to be a novel QTL. The QTL *QFE.ndsu.5B* did not show any association with grain shape and size traits or any other quality traits, suggesting an independent control for FE.

### Lack of *Gpc-B1*, and novel QTL contributed by the non-adapted genotype, offer opportunities to enhance grain protein content (GPC) in adapted wheat germplasm

For GPC, the contribution of the positive alleles for the majority (seven out of nine) of stable QTL by the non-adapted genotype, provides an opportunity to enhance the cultivated gene

**Table 5. Important digenic (QQ) epistatic interactions detected for yield, quality, shape and size related traits using inclusive composite interval mapping in a hexaploid wheat RIL population derived from the cross of an elite genotype (ND 705) and a non-adapted genotype (PI 414566)***.

| Name (QQ) | Trait-Env. | Chr.1 | Pos.1 (cM) | LeftMrk1 | RightMrk1 | Chr.2 | Pos.2 (cM) | LeftMrk2 | RightMrk2 | LOD | PVE (%) | A1 | A2 | A×A |
|---|---|---|---|---|---|---|---|---|---|---|---|---|---|---|
| QQ_NDSU_1A-6B | GH-Carr09 | 1A | 0 | IWB5769 | IWB27821 | 6B | 114 | IWB72400 | IWB9751 | 5.6 | 4.2 | -0.55 | -0.56 | -1.32 |
| QQ_NDSU_1A-6B | GH-Pros09 | 1A | 0 | IWB5769 | IWB27821 | 6B | 114 | IWB72400 | IWB9751 | 6.1 | 3.3 | -0.21 | -0.61 | -1.93 |
| QQ_NDSU_1A-3A2 | DH-Carr09 | 1A | 122 | IWB25443 | IWB3489 | 3A2 | 62 | IWB8177 | IWB33347 | 7.9 | 5.9 | 0.07 | 0.39 | 0.95 |
| QQ_NDSU_1A-3A2 | DH-Pros09 | 1A | 124 | IWB10382 | IWB65643 | 3A2 | 62 | IWB8177 | IWB33347 | 7.0 | 5.1 | -0.11 | 0.86 | 1.61 |
| QQ_NDSU_1A-3A2 | DH-Mean | 1A | 124 | IWB10382 | IWB65643 | 3A2 | 62 | IWB8177 | IWB33347 | 6.8 | 7.7 | -0.05 | 0.20 | 0.90 |
| QQ_NDSU_1A-4A | YLD-Pros10 | 1A | 126 | IWB56353 | IWB40942 | 4A | 104 | IWA4023 | IWB63874 | 5.6 | 4.4 | 17.07 | -38.78 | -222.38 |
| QQ_NDSU_1A-4A | DH-Mean | 1A | 128 | IWA6729 | IWA3019 | 4A | 108 | IWA2606 | IWB8240 | 5.4 | 7.0 | -0.16 | 0.38 | 0.81 |
| QQ_NDSU_1A-6A | KW-Carr10 | 1A | 130 | IWA3019 | IWB45961 | 6A | 180 | IWB22858 | IWB35971 | 5.7 | 10.7 | -0.01 | 0.00 | 0.03 |
| QQ_NDSU_1A-6A | TW-Mean | 1A | 146 | IWB6118 | IWB2101 | 6A | 174 | IWB35951 | IWA4951 | 6.3 | 4.4 | 0.67 | 0.32 | 5.52 |
| QQ_NDSU_1B-6A | KL-Carr09 | 1B | 174 | IWB3934 | IWB36598 | 6A | 114 | IWB3378 | IWB12244 | 5.5 | 9.6 | -0.02 | 0.00 | -0.06 |
| QQ_NDSU_1B-6A | KA-Carr09 | 1B | 182 | IWB20313 | IWB12093 | 6A | 118 | IWB73077 | IWA2018 | 5.1 | 10.5 | -0.05 | -0.03 | -0.16 |
| QQ_NDSU_1D-5A | GH-Pros10 | 1D | 72 | IWB38561 | IWB37574 | 5A | 54 | IWB31441 | IWB6728 | 5.1 | 3.8 | 1.26 | -1.12 | 2.29 |
| QQ_NDSU_1D-5A | GH-Carr10 | 1D | 74 | IWB37574 | IWB60341 | 5A | 70 | IWB40883 | IWB27297 | 5.2 | 7.3 | 0.59 | -0.10 | 1.59 |
| QQ_NDSU_2A-4A | KL-Carr10 | 2A | 0 | IWB71518 | IWB11442 | 4A | 62 | IWB73476 | IWB36420 | 8.0 | 10.8 | 0.04 | -0.01 | -0.07 |
| QQ_NDSU_2A-4A | GPC-Carr09 | 2A | 10 | IWB217 | IWA3122 | 4A | 68 | IWB9590 | IWB47589 | 6.8 | 10.9 | -0.02 | -0.04 | 0.26 |
| QQ_NDSU_2A-7B | GPC-Carr10 | 2A | 86 | IWB63547 | IWB72481 | 7B | 14 | IWB25433 | IWB54918 | 5.2 | 7.3 | -0.11 | -0.09 | 0.34 |
| QQ_NDSU_2A-7B | SPMS-Carr09 | 2A | 88 | IWB6749 | IWB62989 | 7B | 24 | IWB25433 | IWB54918 | 5.8 | 8.1 | -10.62 | -5.50 | 31.78 |
| QQ_NDSU_2B-3A2 | KW-Carr10 | 2B | 54 | IWB47891 | IWB5847 | 3A2 | 62 | IWB8177 | IWB33347 | 5.8 | 10.1 | 0.00 | 0.00 | 0.03 |
| QQ_NDSU_2B-3A2 | KW-Pros10 | 2B | 56 | IWB5847 | IWB32593 | 3A2 | 64 | IWB2973 | IWB54057 | 5.1 | 8.6 | -0.01 | 0.00 | 0.04 |
| QQ_NDSU_2B-2B | DH-Carr09 | 2B | 116 | IWB45403 | IWB7481 | 2B | 154 | IWB44316 | IWA3277 | 7.0 | 5.2 | -0.07 | 0.35 | -1.14 |
| QQ_NDSU_2B-2B | DH-Pros09 | 2B | 116 | IWB45403 | IWB7481 | 2B | 188 | IWB59170 | IWB47236 | 6.2 | 4.3 | -0.10 | 0.59 | -1.90 |
| QQ_NDSU_2B-2B | DH-Carr10 | 2B | 118 | IWB70147 | IWB10568 | 2B | 168 | IWB8334 | IWB44901 | 8.1 | 4.5 | -0.53 | 0.68 | -1.10 |
| QQ_NDSU_2B-2B | DH-Mean | 2B | 124 | IWB7346 | IWB6075 | 2B | 144 | IWB55953 | IWB40021 | 7.6 | 11.1 | -0.93 | 1.12 | -2.06 |
| QQ_NDSU_2B-2B | TW-Pros10 | 2B | 120 | IWB10568 | IWB44618 | 2B | 160 | IWB8294 | IWB10440 | 8.1 | 3.3 | 17.58 | -17.17 | 24.88 |
| QQ_NDSU_2B-2B | TW-Carr09 | 2B | 126 | IWB7346 | IWB6075 | 2B | 154 | IWB44316 | IWA3277 | 6.1 | 6.1 | -1.97 | 1.54 | 5.85 |
| QQ_NDSU_2B-2B | YLD-Pros10 | 2B | 122 | IWB32627 | IWB7346 | 2B | 170 | IWA3621 | IWB49874 | 5.2 | 5.1 | 112.00 | -16.56 | 303.36 |
| QQ_NDSU_2B-2B | FE-Pros09 | 2B | 134 | IWB28615 | IWB7198 | 2B | 146 | IWB40021 | IWB10669 | 5.5 | 10.0 | 1.09 | -1.15 | 1.86 |
| QQ_NDSU_3A2-7A | DH-Carr09 | 3A2 | 38 | IWB23583 | IWB12264 | 7A | 336 | IWB43975 | IWB65210 | 5.4 | 4.3 | 0.26 | 0.06 | 0.81 |
| QQ_NDSU_3A2-7A | GPC-Carr09 | 3A2 | 42 | IWB9039 | IWB27647 | 7A | 350 | IWB11670 | IWB70292 | 5.0 | 8.7 | 0.08 | 0.02 | -0.22 |
| QQ_NDSU_3A2-5A | GH-Pros09 | 3A2 | 144 | IWB10770 | IWB38308 | 5A | 128 | IWB53912 | IWB43972 | 5.6 | 3.5 | -0.11 | 0.64 | -1.93 |
| QQ_NDSU_3A2-5A | GH-Carr09 | 3A2 | 144 | IWB10770 | IWB38308 | 5A | 132 | IWB11865 | IWB2252 | 5.2 | 3.4 | -0.25 | 0.05 | -1.36 |
| QQ_NDSU_3B-4B | GH-Carr09 | 3B | 50 | IWB9212 | IWB30032 | 4B | 94 | IWB7267 | IWB21502 | 5.6 | 3.6 | 0.04 | -0.28 | -1.37 |
| QQ_NDSU_3B-4B | GH-Pros10 | 3B | 54 | IWB24336 | IWB7629 | 4B | 106 | IWA908 | IWB1606 | 5.5 | 3.9 | 0.71 | 1.27 | -2.19 |
| QQ_NDSU_3D2-3D2 | FE-Carr10 | 3D2 | 2 | IWB28189 | IWB7136 | 3D2 | 10 | IWB56509 | IWB24277 | 5.8 | 2.2 | -4.23 | 3.87 | 3.79 |
| QQ_NDSU_3D2-3D2 | TW-Pros10 | 3D2 | 8 | IWB50521 | IWB5600 | 3D2 | 18 | IWB12046 | IWB10466 | 6.8 | 5.0 | 20.55 | -20.22 | 34.08 |
| QQ_NDSU_4A-5A | DH-Carr09 | 4A | 204 | IWB6183 | IWB57472 | 5A | 332 | IWB38165 | IWB235 | 5.1 | 4.6 | 0.15 | -0.15 | 0.85 |
| QQ_NDSU_4A-5A | DH-Pros09 | 4A | 206 | IWB29018 | IWB29438 | 5A | 334 | IWB38165 | IWB235 | 6.7 | 6.1 | 0.19 | -0.50 | 1.84 |

*(Continued)*

**Table 5.** (*Continued*)

| Name (QQ) | Trait-Env. | Chr.1 | Pos.1 (cM) | LeftMrk1 | RightMrk1 | Chr.2 | Pos.2 (cM) | LeftMrk2 | RightMrk2 | LOD | PVE (%) | A1 | A2 | A×A |
|---|---|---|---|---|---|---|---|---|---|---|---|---|---|---|
| QQ_NDSU_4A-6A | GH-Pros09 | 4A | 270 | IWB34615 | IWB11801 | 6A | 88 | IWB58333 | IWB73827 | 5.1 | 2.7 | -0.40 | 0.18 | 1.91 |
| QQ_NDSU_4A-6A | GH-Pros10 | 4A | 290 | IWB24691 | IWB6186 | 6A | 84 | IWB5521 | IWB72039 | 6.1 | 3.7 | -0.28 | 0.08 | 2.51 |
| QQ_NDSU_4A-4A | FE-Mean | 4A | 294 | IWA3192 | IWB57180 | 4A | 310 | IWB71698 | IWB4670 | 5.3 | 3.0 | -0.91 | 1.09 | 1.28 |
| QQ_NDSU_4A-4A | FE-Carr10 | 4A | 306 | IWB55374 | IWB71700 | 4A | 308 | IWB9276 | IWB29950 | 8.8 | 2.4 | 2.64 | -2.65 | 3.31 |
| QQ_NDSU_4B-7B | KL-Carr09 | 4B | 90 | IWB11611 | IWB9202 | 7B | 0 | IWB1438 | IWB27109 | 6.1 | 9.8 | 0.01 | 0.02 | -0.06 |
| QQ_NDSU_4B-7B | KL-Mean | 4B | 90 | IWB11611 | IWB9202 | 7B | 0 | IWB1438 | IWB27109 | 5.1 | 10.2 | 0.00 | 0.02 | -0.03 |
| QQ_NDSU_5A-5A | DH-Pros10 | 5A | 206 | IWB8905 | IWB27060 | 5A | 258 | IWB35391 | IWB64718 | 6.6 | 12.0 | -1.82 | -0.92 | 1.69 |
| QQ_NDSU_5A-5A | DH-Mean | 5A | 206 | IWB8905 | IWB27060 | 5A | 258 | IWB35391 | IWB64718 | 6.0 | 27.9 | -1.20 | -0.63 | 1.17 |
| QQ_NDSU_5B-7A | GH-Pros09 | 5B | 56 | IWB8553 | IWB27245 | 7A | 228 | IWB34649 | IWB54262 | 7.3 | 4.4 | 0.54 | 0.53 | -2.40 |
| QQ_NDSU_5B-7A | GH-Carr09 | 5B | 58 | IWB8553 | IWB27245 | 7A | 228 | IWB34649 | IWB54262 | 5.2 | 4.5 | 0.26 | 0.75 | -1.59 |
| QQ_NDSU_5B-7A | GH-Pros10 | 5B | 74 | IWA1776 | IWB71913 | 7A | 226 | IWB30703 | IWA4601 | 5.2 | 3.0 | 0.21 | 0.56 | -2.25 |
| QQ_NDSU_6D3-6D3 | TW-Pros10 | 6D3 | 4 | IWB44754 | IWB49496 | 6D3 | 12 | IWB44239 | IWB7372 | 5.1 | 2.6 | 28.40 | -30.47 | 33.82 |
| QQ_NDSU_6D3-6D3 | FE-Carr10 | 6D3 | 8 | IWB10505 | IWB505 | 6D3 | 10 | IWB38232 | IWB44239 | 5.7 | 2.4 | 2.80 | -2.69 | 3.21 |

FE = Flour extraction (%), GPC = Grain protein content (%), GH = Grain hardness (HI), GY = Grain yield (kg ha-1), SPMS = No of spikes m2, GPS = No of Grains per spike, DH = Days to heading (d), PH = Plant height (cm), GH = Grain length, GW = Grain width, GA = Grain area, GLWR = Grain length width ratio, TGW = Thousand grain weight, TW = Test weight

*Chr. = Chromosome; Pos. = Position; Mrk. = Marker; A = Additive effect

pool for increasing GPC in wheat. The major and stable QTL *QGPC.ndsu.5B* (5BS), *QGPC.ndsu.7A.2* (7AL) and *QGPC.ndsu.7B* (7BS) could be of significant interest as these QTL were independent of grain yield, as well as grain shape and size related traits, and PI 414566 contributed the positive alleles. These independent QTL could be useful to enhance GPC through MAS, without compromising yield. Comparison with 49 GPC studies (reviewed in Kumar et al. [12]) suggests that *QGPC.ndsu.5B* (located on 5BS) and *QGPC.ndsu.7A.2* (located on 7AL) could be novel QTL present in the non-adapted germplasm. Although El-Feki et al. [44] identified a QTL on 5BS, that QTL was located too far from the location of *QGPC.ndsu.5B*. Stable QTL for GPC on 7AL were also reported in few earlier studies in both durum [45, 46] and hexaploid wheat [47, 48]. The QTL *QGPC.ndsu.7A.2* was located near the telomeric end of 7AL, but the QTL reported earlier [45–48] were located in the middle of the chromosome arm 7AL. A QTL in the telomeric region of 7BS, where *QGPC.ndsu.7B* was identified, have also been reported in both tetraploid [49–51] and hexaploid wheat [52, 53]. The earlier studies by Blanco et al. [49, 50] and Bogard et al. [53], however, found a negatively associated QTL for yield or yield components in this region as well. In contrast, the QTL identified in this study did not show any genomic association with GY, GPS, SPMS, GL, GW, GA and TGW. This could be because of differences in germplasm and environmental conditions. We could not find any QTL on 6BS, suggesting that *GPC-B1*, the major GPC QTL cloned in wheat [54], does not play a role in our germplasm. This is expected as the majority of the modern wheat varieties carry the non-functional allele of *Gpc-B1* gene [54]. The cloning of *Gpc-B1* has now facilitated the successful introgression of a functional copy of this gene to adapted germplasm, through MAS ([55]; for review see, Kumar et al. [12]). This suggests an opportunity to further improve GPC in our spring wheat germplasm, through introgression of the high GPC allele of *Gpc-B1* gene.

## Novel QTL for grain hardness on chromosome 4B

Grain hardness or grain texture has a direct relationship with important end use quality traits like milling yield or flour extraction in wheat. This was further confirmed in our study, where we also observed a moderate positive association of GH with flour extraction in all the environments. Many studies have shown that grain hardness in wheat is controlled by a major hardness locus (*Ha*) located at a sub-telomeric position on chromosome 5DS [56, 57]. The *Ha* locus also harbors genes encoding 15-kD lipid binding endosperm specific protein, called friabilins. The friabilins are composed of mainly two proteins puroindoline a (Pina) and puroindoline b (Pinb) [58]. Characterization of different wheat types have shown that hard wheat varieties either lack or carry specific mutations for the puroindoline coding genes [57]. The soft wheat varieties carry the wild-type puroindoline alleles [59]. In addition to the role of *Ha* locus, studies in the past have also reported several other QTL associated with hardness as well [16, 60–63]. We did not find any QTL for GH on 5DS, suggesting that both the parental genotypes probably carry the hardness alleles at *Ha* locus. In this population, the differences in GH are attributed to several other QTL, which were mostly detected in multiple environments, showing a greater influence of genotype than environment on the GH, as has been observed in the past [64]. The QTL *QGH.ndsu.1A*, *QGH.ndsu.1B*, and *QGH.ndsu.4B.1* could be considered more important for GH due to their stability across environments and significant additive effect. The QTL *QGH.ndsu.1B*, and *QGH.ndsu.5A.2* are in the same genomic region where Heo and Sherman [63] identified a major QTL for GH. Similar to the location of *QGH.ndsu.1A*, Groos et al. [62] also reported a QTL for hardness in the telomeric region of 1A. We could not find any study reporting a QTL on 4B, suggesting that *QGH.ndsu.4B.1* could be a novel QTL. The contribution of hardness alleles by non-adapted parent for the two major loci *QGH.ndsu.1A* and *QGH.ndsu.1B* provides an opportunity to incorporate those desirable loci into the breeding program.

## Complex genetics of yield and component traits

Grains per spike (GPS), number of spikes per square meter (SPMS) and thousand grain weight (TGW) are considered the three most important components of yield in wheat. The phenotypic structure in our study showed similar results, where GPS, SPMS and TGW had low to moderate positive correlations (mostly significant) with GY. However, yield is one of the most complex traits which is highly influenced by the environment. The genetic dissection further confirms this, as most of the identified QTL were environment specific and did not show stability across locations. The genomic region 17 (3D), where ND705 alleles were associated with increased GY (*QGY.ndsu.3D*), GPS (*QGPS.ndsu.3D*) and SPMS (*QSPMS.ndsu.3D*), and decreased PH (*QPH.ndsu.3D*), is a good candidate for yield increase without any effect on GPC. However, ND705 alleles at this locus were associated with decrease in TGW and GA [22]. This could probably mean that the yield increase through this locus is a result of increase in grains per spike and spikes per square meter. The genomic region 19 (4B) has been found associated with significant effect on grain shape and size, not only in wheat (discussed in detail by Kumar et al. [22]), but in other cereals as well [65–67]. In this population, the genomic region 19 was associated with major and stable effect on TGW, GL, GLWR,GA and GW [22]. We observed that the same region is also associated with GY, SPMS and GPC. For this locus, the ND 705 alleles, although increased GY, SPMS, GPC and TW, they were associated with decrease in other components, suggesting that the yield increase is due to an increase in SPMS. Due to its expression across environments, *QGPS.ndsu.2B* (on 2BL) associated with grains per spike could be an important candidate for yield improvement. Many other studies in the past have also identified QTL for grains per spike on chromosome 2B [68–71]. However,

comparison of map locations showed that those QTL were located on the short arm, whereas *QGPS.ndsu.2B* was located in the centromeric bin of 2BL, suggesting that this is probably a novel QTL specific to our germplasm. Lines with ND alleles at both *QGPS.ndsu.2B* and *QGPS.ndsu.3D* had on average three more grains per spike (36.1 *vs* 33.1) compared to the lines with PI 414566 alleles at both loci, suggesting the potential of these two QTL for yield improvement.

## Heading date controlled by major loci in the genomic regions harboring *VRN* genes on group 5 chromosomes

The photoperiod (*Ppd*) response, vernalization (*Vrn*) and earliness *per se* (*Eps*) genes control flowering time in wheat. The *Eps* genes influence flowering time independently, whereas, *Ppd* and *Vrn* genes have a modifying effect. To date, several QTL/genes for heading date are reported in wheat [72]. The most significant loci associated with photoperiod response, *Ppd-D1*, an orthologous to the *Arabidopsis* photoperiod pathways gene *CONSTANS* [73], was mapped on chromosome 2D in wheat [74]. The photoperiod insensitive allele of this gene *Ppd-D1a* was widely used in the "green revolution". We did not find any loci for HD on chromosome 2D, suggesting that the major developmental gene *Ppd-D1* did not play any role in the variation for heading date in our population. This could also probably mean that both parents have the same alleles for this gene. The quantitative variation for heading date in our population was due to stable loci on 5AL and 5BL. Comparison with earlier studies suggest that *QDH.ndsu.5A* and *QDH.ndsu.5B* are in the region of *VRN-A1* and *VRN-B1*, respectively [37, 75]. This is probably expected as the parental genotype PI 414566 is a facultative type (or winter type) wheat. In contrast to Guedira et al. [75], we observed that *QDH.ndsu.5A* has a significantly higher contribution compared to *QDH.ndsu.5B* in our population. This could be attributed to differences in germplasm.

## Major locus for plant height in the region of *Rht8* on chromosome arm 2DS

Plant height affects lodging, disease resistance [76], quality [77] and yield [78] in wheat and therefore, has been an important breeding target since the mid-nineteenth century [79]. The reduced height phenotype which revolutionized the wheat yield worldwide could be mainly attributed to the Japanese varieties 'Norin 10' carrying the *Rht-B1b* (or *Rht1* on chromosome 4B), *Rht-D1b* (or *Rht2* on chromosome 4D) genes and 'Akakomugi' carrying the *Rht8* (on chromosome 2D) gene [79, 80]. In addition, over the years, many other reduced height or dwarfing genes/QTL have been identified in wheat [21, 69, 81]. In our population, although both the cultivars were semi-dwarf and differed significantly for PH (ND705 = 95cm, PI 414566 = 70 cm), the major reduced height genes *Rht-B1b* and *Rht-D1b* did not play any role in the variation for PH. The QTL *QPH.ndsu.2D*, which was identified in the region of the *Rht8* gene [82], played larger role in controlling PH. The comparison of map locations suggests that *QPH.ndsu.2D* is most likely *Rht8*. The two groups of lines with ND705 and PI 414566 alleles at the *QPH.ndsu.2D* locus showed that the PI 414566 alleles significantly reduced plant height.

The semi-dwarfed wheat genotypes show improved spikelet fertility and increased grain number per spike compared to the tall genotypes because they distribute a major portion of assimilate to the developing spikes [83]. The *Rht8* locus has been found associated with yield and several yield components including spike length, spikelet number, spikelet compactness, and thousand grain weight [2, 68, 81, 82, 84]. In this population, we did not find any QTL associated with yield, yield components, grain shape and size related traits in the *Rht8* region. However, comparison of groups with ND705 or PI 414566 alleles at the *Rht8* locus showed that the reduced height allele from the PI 414566 at the *Rht8* locus was associated with a significant increase in grains per spike and flour extraction. But, the same allele also showed a

decrease in the number of spikes per square meter and test weight. No significant association of the reduced height allele at the *Rht8* was observed with grain yield, or other component traits, including thousand grain weight, grain length, grain width, grain length width ratio and grain area.

According to some estimates, over 70% of the registered wheat cultivars in the world are fixed for the *Rht-B1b* and the *Rht-D1b* dwarfing genes [85]. Since Norin 10 was the source of introgression of reduced height to US genotypes [79, 80], *Rht-B1b* and *Rht-D1b* are much more prevalent in the US wheats, while the frequency of the *Rht8* is much lower [86]. On the other hand, the reduced height in the European and Asian cultivars was contributed initially by the *Rht8* gene using the Japanese cultivar Akakomugi [79]. In a Chinese wheat collection, Zhang et al. [87] observed that the frequency of *Rht8* was highest (46.8%), followed by *Rht-D1b* (45.5%) and *Rht-B1b* (24.5%). Since PI 414566 was originally from China, it is more likely that it harbors the reduced height allele *Rht8c*, in addition to either the *Rht-B1b* or *Rht-D1b*. Overall, the historical evidence and the genotypic and phenotypic structure suggests that most likely, either the *Rht-B1b* or *Rht-D1b* locus is fixed in both the parental genotypes of our population, while the variation for PH was contributed mainly by the *Rht8*. The availability of molecular markers, now provides the opportunity to exploit the benefits of the *Rht8* in US wheats. Introgression and selection of gibberellins insensitive genes *Rht-B1b* or *Rht-D1b* with gibberellins insensitive gene *Rht8* using MAS may facilitate rapid development of high-yielding varieties [88].

## Association of grain shape and size with wheat quality and yield

The knowledge about the genetic association of grain shape and size traits with grain quality, particularly flour extraction or milling yield in wheat is, although important, but mostly missing. The phenotypic and genotypic architecture of our population suggested that grain shape and size traits (GL, GW, GA, TGW and TW) were although correlated with quality traits, the association was not strong. The fact that only ~18% of the loci for quality share the genomic locations with loci for grain shape and size, suggested that the genetic control of wheat quality traits like protein content and flour extraction is mostly independent of grain shape and size.

Although theoretical studies suggested that large, spherical grains are optimal for milling yield, presumably due to increased endosperm [89], experimental studies based on phenotypic evaluations did not find a strong correlation between flour extraction and GL, GW, GA, TGW, TW [90, 91]. Similarly, in our study, flour extraction was not correlated with GL in any of the environments. However, a significant positive but weak correlation between GW and FE in Prosper location suggests that the association is probably due to environmental factors and not because of genetic loci. This was further confirmed by genetic analysis, which showed that the loci for FE do not share any genomic region with grain shape and size.

As expected, grain shape and size traits had more significant effect on yield compared to their effect on quality in wheat. The observation that about half of the genomic regions for yield traits were also associated with grain shape and size suggests a significant role of grain morphology in enhancing yield in wheat. The genomic regions for GA and TGW were most commonly associated with yield or yield traits, thus confirming that TGW is one of the most important yield components in wheat. Since GA is highly correlated with TKW ([22]; Fig 2), targeting an increase in GA in the breeding programs may help enhancing wheat yield through an increase in TGW.

## Role of epistasis in the genetic control of quantitative traits

Under the effect of epistasis, the value of an allele or genotype at a locus depends on the genotype at other epistatically interacting loci, thus complicating the picture of gene action. A

seemingly "favorable" allele at one locus may be an "unfavorable" allele in a different genetic background. Knowledge about the role of epistasis in the genetic control of target traits is crucial, because epistasis complicates the genotype–phenotype relationship, and thus affects the rate of genetic gain in plant breeding. In this study, a comparable PV explained by main effect and epistasis (41.8% & 48.2%, respectively) clearly suggests that in addition to main effects, epistasis plays an important role in the genetic architecture of most of the quantitative traits. Also, the observation that almost all the QTL involved in epistasis did not have any effect of their own, but contributed to the genetic variation of the traits purely through epistatic interactions, suggests the need to use statistical methods/software which have the ability to detect such QTL.

The role of epistasis in the genetic control of quantitative traits has been documented in many crops [92], including wheat [1, 70, 71], barley [93], maize [94], and rice [95, 96]. For grain protein content in two RIL populations, Kulwal et al. [1] reported that epistatic interactions contributed a significantly higher portion of phenotypic variation than main effect QTL. Similar results were also reported for yield related traits in wheat [97]. From these studies, we can conclude that epistatic interactions are an important component of the genetic basis of most of the quantitative traits. Neglecting epistasis could lead to reduced selection response and thus need serious consideration in breeding for genetic improvement. In the past, despite the identification of epistatic interactions in different crops and the success in marker assisted selection, epistatic QTL were hardly employed in genetic improvement programs. This could be due to many reasons, including the complexity of the epistatic gene networks and minor effects contributed by individual interactions. This means if we have to achieve greater selection response through the application of genomic tools, we need to develop improved tools to understand the role of epistatic interactions and their incorporation into the breeding program. The methods of genomic selection [98] might help us to incorporate epistasis into the breeding pipeline to achieve the desired genetic gains in crops.

## Implications of the current study for wheat improvement

Wheat quality and yield improvement are two important target areas of any wheat breeding program worldwide. These traits have complex genetic control and low to moderate heritability. Here, the high resolution genetic dissection provided tightly linked molecular markers for quality and yield that could expedite the process of development of improved varieties in spring wheat. This study not only enhanced our knowledge about the genetics of wheat quality and yield, but also provided new detailed evidence about the phenotypic and genetic association of wheat quality and yield with grain shape and size related traits. The quality traits showed mostly independent genetic control whereas, yield traits showed moderate genetic association with grain shape and size traits. Based on these observations, it could be suggested that targeting grain shape and size, particularly an increase in GA, may help enhancing wheat yield through an increase in TGW. The novel and yield independent QTL for GPC identified in this study offer the opportunity to improve GPC in hard red spring wheat germplasm, without any yield loss (Table 3). Five lines (RIL No: 4, 10, 40, 90, and 130) with the highest GPC combine all the positive alleles from both the parents at the eight stable loci for this trait, providing important pre-breeding material to enhance protein content in the adapted germplasm. When all positive alleles were present together, these lines showed about two LSD higher GPC compared to the parental genotypes (or about 12% higher compared to ND 705 alleles) (S5 Table). Another line (RIL-138), which combines the allele from PI 414566 at *QGPC.ndsu.7A.2* with other alleles from ND 705, showed almost 1.5 LSD higher GPC and no significant loss in grain yield when compared to the elite line ND 705 (S5 Table). This line could also serve as

valuable pre-breeding germplasm to enhance protein content without compromising grain yield. Similarly, alleles from both the parents at the stable loci could be combined to increase flour extraction in wheat (Table 3). The novel and consistent QTL for grains per spike identified on chromosome 2B and for flour extraction on chromosome 5B need to be incorporated into wheat breeding programs and could be very useful in enhancing grain and milling yield, respectively.

About 32% of the positive QTL alleles were contributed by PI 414566, explaining the transgressive segregation in the population, as a result of combination of superior alleles. The unadapted line PI 414566 also contributed novel, stable and major loci for FE, GPC, GH, and PH, offering an opportunity to enhance the adapted gene pool for these important traits. This further confirmed that non-adapted germplasm is essential to enhance genetic diversity in cultivated germplasm. The RILs having combinations of positive alleles from both the parents at important loci could serve as important pre-breeding material for improvement of target quality and yield traits. In conclusion, the genomic resources developed in this study will facilitate marker assisted breeding efforts for wheat improvement.

## Supporting information

**S1 Table. Genetic maps of ND 705 × PI 414566 RIL population showing the location of unique loci.** Complete genetic map is available in Kumar et al. [22].
(XLSX)

**S2 Table. Phenotypic performance of elite genotype ND705, non-adapted genotype PI 414566, and their recombinant inbred lines (RILs) in different environments.**
(XLSX)

**S3 Table. Genomic (QTL) regions associated with grain quality and yield related traits in an RIL population derived from the cross of an elite genotype (ND 705) and a non-adapted genotype (PI 414566).**
(XLSX)

**S4 Table. Digenic epistatic interactions for 14 different traits related to yield, quality, grain shape and size in bread wheat identified using QTL IciMapping.**
(XLSX)

**S5 Table. Allelic composition of few selected recombinant inbred lines (RILs) at grain protein content QTL identified in this study.**
(XLSX)

## Acknowledgments

The authors would also like to thank J. Underdahl, M. Abdallah, A. Walz and other members of the hard red spring wheat project and wheat quality lab for their excellent technical assistance.

## Author Contributions

**Conceptualization:** Ajay Kumar, Mohamed Mergoum.

**Data curation:** Ajay Kumar, Eder E. Mantovani, Senay Simsek.

**Formal analysis:** Ajay Kumar, Eder E. Mantovani, Shalu Jain, Elias M. Elias.

**Funding acquisition:** Mohamed Mergoum.

**Investigation:** Ajay Kumar.

**Methodology:** Ajay Kumar.

**Project administration:** Ajay Kumar, Mohamed Mergoum.

**Resources:** Ajay Kumar, Mohamed Mergoum.

**Supervision:** Ajay Kumar, Mohamed Mergoum.

**Writing – original draft:** Ajay Kumar.

**Writing – review & editing:** Senay Simsek, Shalu Jain, Elias M. Elias.

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
