## [Decision Letter · Decision Letter 0]

26 Jun 2019

PONE-D-19-14666

Genome wide genetic dissection of wheat quality and yield related traits and their relationship with grain shape and size traits in an elite × non-adapted bread wheat cross

PLOS ONE

Dear Dr. Kumar,

Thank you for submitting your manuscript to PLOS ONE. After careful consideration, we feel that it has merit but does not fully meet PLOS ONE’s publication criteria as it currently stands. Therefore, we invite you to submit a revised version of the manuscript that addresses the points raised during the review process.

We would appreciate receiving your revised manuscript by Aug 10 2019 11:59PM. To enhance the reproducibility of your results, we recommend that if applicable you deposit your laboratory protocols in protocols.io, where a protocol can be assigned its own identifier (DOI) such that it can be cited independently in the future. For instructions see: http://journals.plos.org/plosone/s/submission-guidelines#loc-laboratory-protocols

We look forward to receiving your revised manuscript.

Kind regards,

Aimin Zhang, Ph.D.

Academic Editor

PLOS ONE

Journal Requirements:

Reviewers' comments:

Reviewer's Responses to Questions

**Comments to the Author**

1. Is the manuscript technically sound, and do the data support the conclusions?

Reviewer #1: Yes

Reviewer #2: Yes

2. Has the statistical analysis been performed appropriately and rigorously? 

Reviewer #1: Yes

Reviewer #2: Yes

3. Have the authors made all data underlying the findings in their manuscript fully available?

Reviewer #1: Yes

Reviewer #2: Yes

4. Is the manuscript presented in an intelligible fashion and written in standard English?

Reviewer #1: Yes

Reviewer #2: Yes

5. Review Comments to the Author

Reviewer #1: The following expresses my opinion about the merits of the study:

1) The paper provides useful information for breeders of hard red spring wheat and for the wider wheat community, particularly since the paper addresses the association of wheat quality and yield traits with grain shape and size traits, a poorly studied area.

2) The use of a high-density map for the mapping procedures should enable high efficiency application of MAS in future breeding programmes based upon this germplasm or upon these general results.

3) The identification of novel alleles provided by the non-adapted line, especially (i) alleles for improved GPC with no detrimental effects on yield or other traits, and (ii) alleles influencing grain hardness, together potentially provide new opportunities for achieving genetic progress in these traits.

4) Other valuable features of the paper are: (i) the detailed analysis of epistasis, which provides additional evidence of the importance of taking this into account during selection; (ii) the evidence presented that selection for grain area during breeding might also provide dividends; and (iii) the extensive comparison between the QTLs mapped here and those published previously, which details which of the QTLs identifies here are truly novel.

A comment: the study would probably have been more valuable still if more than 160 RILs had been evaluated and, to boot, more site-year combinations had been included; nonetheless, I do not see these drawbacks as sufficient motive to hold up publication of the study as it stands.

Am I right in thinking that there is disagreement in the labelling of the parental phenotypic mean values between Table 1 and Fig. 1? I recommend checking this.

Grammatically the writing is generally clear, although it might be further improved by being looked over once more.

Based upon the above opinion, I feel the paper is well worthy of publication.

Reviewer #2: The manuscript has some new finding in the area (specified for hard red spring wheat) which can be useful in the breeding program.

It has dissected the genetics of wheat quality and yield traits and also identified novel QTL and markers closely associated with quality and yield. It shows data about the epistasis in the genetic control of quality and yield as well.

The manuscript is well written. My only concern is that only two replicates were used in this experiment. I think it should be more.

I think it is suitable for publication.

6. PLOS authors have the option to publish the peer review history of their article (what does this mean?). If published, this will include your full peer review and any attached files.

Reviewer #1: No

Reviewer #2: No

---

## [Author Response · Author response to Decision Letter 0]

2 Jul 2019

Response to reviewer’s comments

Journal Requirements:

Author’s response: We have formatted the manuscript according to PLOS One style and hope that it meet the journal requirements. 

Author’s response: The phrase “data not shown” was removed from the manuscript.

Reviewers' comments:

Reviewer's Responses to Questions

Comments to the Author

1. Is the manuscript technically sound, and do the data support the conclusions?

Reviewer #1: Yes

Reviewer #2: Yes

2. Has the statistical analysis been performed appropriately and rigorously? 

Reviewer #1: Yes

Reviewer #2: Yes

3. Have the authors made all data underlying the findings in their manuscript fully available?

Reviewer #1: Yes

Reviewer #2: Yes

4. Is the manuscript presented in an intelligible fashion and written in standard English?

Reviewer #1: Yes

Reviewer #2: Yes

5. Review Comments to the Author

Reviewer #1: The following expresses my opinion about the merits of the study:

1) The paper provides useful information for breeders of hard red spring wheat and for the wider wheat community, particularly since the paper addresses the association of wheat quality and yield traits with grain shape and size traits, a poorly studied area.

2) The use of a high-density map for the mapping procedures should enable high efficiency application of MAS in future breeding programmes based upon this germplasm or upon these general results.

3) The identification of novel alleles provided by the non-adapted line, especially (i) alleles for improved GPC with no detrimental effects on yield or other traits, and (ii) alleles influencing grain hardness, together potentially provide new opportunities for achieving genetic progress in these traits.

4) Other valuable features of the paper are: (i) the detailed analysis of epistasis, which provides additional evidence of the importance of taking this into account during selection; (ii) the evidence presented that selection for grain area during breeding might also provide dividends; and (iii) the extensive comparison between the QTLs mapped here and those published previously, which details which of the QTLs identifies here are truly novel.

Author’s response: Thank you so much for your comments, wonderful observations about our manuscript and your recommendation for publication. 

A comment: the study would probably have been more valuable still if more than 160 RILs had been evaluated and, to boot, more site-year combinations had been included; nonetheless, I do not see these drawbacks as sufficient motive to hold up publication of the study as it stands.

Author’s response: Thank you so much for your comment. We understand that larger sample size and more site-year combinations, may provide additional information. Larger population may help identify some additional minor QTL (if any). That being said, we strongly feel that (and as the results indicate), this study has identified important QTL with significant effects. In QTL mapping studies, it is a common and acceptable practice to have a population of >100 and minimum of three site-year combinations (our study used data for 160 RILs from four site-year combinations). One of the main reason for this is the large amount of resources required to carry out field evaluation at multiple locations. Moreover, the phenotypic assays for evaluation of quality traits are labor intensive and require huge amount of resources, thus limiting site-year combinations to a manageable and acceptable number. 

Am I right in thinking that there is disagreement in the labelling of the parental phenotypic mean values between Table 1 and Fig. 1? I recommend checking this.

Author’s response: Thank you so much for noticing this mistake. We have now made necessary corrections to the legend of Figure 1. 

Grammatically the writing is generally clear, although it might be further improved by being looked over once more.

Author’s response: Thank you for your comment. As can be seen in the manuscript, we have gone through the manuscript one more time to improve the manuscript. 

Based upon the above opinion, I feel the paper is well worthy of publication.

Reviewer #2: The manuscript has some new finding in the area (specified for hard red spring wheat) which can be useful in the breeding program.

It has dissected the genetics of wheat quality and yield traits and also identified novel QTL and markers closely associated with quality and yield. It shows data about the epistasis in the genetic control of quality and yield as well.

The manuscript is well written. My only concern is that only two replicates were used in this experiment. I think it should be more.

I think it is suitable for publication.

Author’s response: Thank you so much for your review of our manuscript and your recommendation. 

We understand that more replications are always better, but due to resource and labor intensive phenotyping in multi-environment mapping studies, it is a standard and acceptable practice to have two replicates per environment. Moreover, the phenotypic assays for evaluation of quality traits are highly labor intensive and require huge amount of resources, thus limiting number of replicates and site-year combinations to a manageable and acceptable number.

---

## [Decision Letter · Decision Letter 1]

22 Jul 2019

PONE-D-19-14666R1

Genome wide genetic dissection of wheat quality and yield related traits and their relationship with grain shape and size traits in an elite × non-adapted bread wheat cross

PLOS ONE

Dear Dr. Kumar,

Thank you for submitting your manuscript to PLOS ONE. After careful consideration, we feel that it has merit but does not fully meet PLOS ONE’s publication criteria as it currently stands. Therefore, we invite you to submit a revised version of the manuscript that addresses the points raised during the review process.

We would appreciate receiving your revised manuscript by Sep 05 2019 11:59PM. To enhance the reproducibility of your results, we recommend that if applicable you deposit your laboratory protocols in protocols.io, where a protocol can be assigned its own identifier (DOI) such that it can be cited independently in the future. For instructions see: http://journals.plos.org/plosone/s/submission-guidelines#loc-laboratory-protocols

We look forward to receiving your revised manuscript.

Kind regards,

Aimin Zhang, Ph.D.

Academic Editor

PLOS ONE

Reviewers' comments:

Reviewer's Responses to Questions

**Comments to the Author**

1. If the authors have adequately addressed your comments raised in a previous round of review and you feel that this manuscript is now acceptable for publication, you may indicate that here to bypass the “Comments to the Author” section, enter your conflict of interest statement in the “Confidential to Editor” section, and submit your "Accept" recommendation.

Reviewer #1: (No Response)

2. Is the manuscript technically sound, and do the data support the conclusions?

Reviewer #1: Yes

3. Has the statistical analysis been performed appropriately and rigorously? 

Reviewer #1: Yes

4. Have the authors made all data underlying the findings in their manuscript fully available?

Reviewer #1: Yes

5. Is the manuscript presented in an intelligible fashion and written in standard English?

Reviewer #1: Yes

6. Review Comments to the Author

Reviewer #1: As requested, the authors have (i) adequately defended the questions posed on sample size and the number of site-year combinations included; (ii) corrected the discrepancy between Table 1 and Fig. 1; and (iii) improved the grammar.

I therefore maintain my original opinion that the paper is well worthy of publication.

Nonetheless, below I include additional modifications to the grammar, those that I have managed to identify in the time available:

Personally, I would generally replace “QTL” with “QTLs” where it is used in the plural, but do not insist since papers vary in this respect.

The following are more specific modifications suggested to particular lines in the manuscript:

l. 56: insert “the” after “while”.

l. 59: insert “it” after “although”.

l. 60: replace “they do” with “does”.

l. 61: insert “with them” after “loci”, insert “A” before “major”, insert “the” before “Rht8” and insert “the” before “reduced”.

l. 62: replace “increase” with “increases”.

l. 97: replace “more” with “a greater”.

l. 259: insert “The” before “same”.

l. 277: replace “negatively” with “negative”.

l. 298: replace “has” with “had”.

l. 299: replace “effect” with “effects”.

l. 314: replace the phrase with “The majority (52 out of 60) of these QTL”.

l. 362: insert “the” before “PI”.

l. 363: insert “the” before “PI”.

l. 364: insert “the” before “ND”.

l. 415: replace “,” with “and” and insert “,” after “SPMS”.

l. 420: replace “had association” with “were associated”.

l. 432: replace “those” with “these”.

l. 434: insert “the” before “PI” and replace “were” with “was”.

l. 440: replace “interaction” with “interactions”.

l. 444: replace “More” with “A greater”.

l. 455: delete “each” and insert “for each trait” after “detected”.

l. 456: replace “locus” with “loci”.

l. 457: replace “interaction” with “interactions”.

l. 458: delete “or”.

l. 473: replace “helps” with “help” and replace “marker” with “market”.

l. 484: insert “a” after “of”.

l. 491: insert “,” after “effect”.

l. 494: insert “the” after “that”, replace “increase” with “increased” and delete “Like this study”.

l. 495: delete “as well”.

l. 500: insert “the” after “by” and insert “,” after “genotype”.

l. 502: insert “the” before “contribution” and “the” before “majority”.

l. 505: insert “,” after “yield”.

l. 506: insert “,” after “traits”.

l. 508: delete “,”.

l. 510: delete “,” after “Although”.

l. 513: delete “,” before “were”.

l. 515: replace “have” with “has”.

l. 516: insert “,” after “however”.

l. 519: replace “difference” with “differences”.

l. 520: insert “,” before “does”.

l. 521: insert “the” after “as”.

l. 522: insert “the” after “allele of”.

l. 523: insert “a” before “functional”.

l. 525: insert “the” before “high” and insert “the” before “Gpc-B1”.

l. 529: insert “,” before “where”.

l. 531/2: delete “which is”.

l. 534: insert “,” after “proteins”.

l. 537: insert “the” after “of”.

l. 538: delete “as well”.

l. 539: delete “,” after “genotypes”.

l. 541: replace “major” with “greater”.

l. 553: delete “,” after “(SPMS)”.

l. 554: insert “the” after “three”.

l. 556: replace “trait” with “traits”, insert “,” after “traits” and delete “which is”.

l. 560: insert “,” before “is”.

l. 564: replace “effect” with “effects”.

l. 566: replace “effect” with “effects”.

l. 568: insert “,” after “alleles”, replace “increase” with “they increased” and delete “they”.

l. 569: replace “decrease” with “decreases” and insert “,” after “components”.

l. 570: replace “environment” with “environments”.

l. 575: replace “an” with “on”.

l. 583: replace “Till” with “To”.

l. 585: insert “the” after “to”.

l. 586: insert “the” after “in”.

l. 587: insert “,” after “2D”.

l. 588: replace “papulation” with “population”.

l. 589: insert “the” before “alleles”.

l. 599: insert “the” after “since”.

l. 601: insert “the” before “Japonese”.

l. 604: delete “,” after “although” and replace “differ” with “differed”.

l. 607: insert “the” before “Rht8” and insert “a” before “larger”.

l. 608: replace “suggest” with “suggests”.

l. 609: insert “the” after “at”.

l. 617: insert “the” after “at”.

l. 618: insert “the” after “at”.

l. 619: delete “,”, insert “a” after “showed” and insert “the” after “in”.

l. 620: insert “the” after “of”.

l. 623: insert “,” after “estimates”.

l. 626: insert “the” after “while”.

l. 627: insert “the” after “by” and insert “the” after “using”.

l. 628: delete “,” after “[86]”.

l. 629: delete “,” after “Since”.

l. 630: insert “the” after “harbors” and replace “alleles” with “allele”.

l. 632: insert “the” after “either” and replace “loci” with “locus”.

l. 634: delete “,”.

l. 635: insert “the” after “of” and replace “gene” with “genes”.

l. 641: insert “,” after “is”, insert “,” after “important” and delete “but”.

l. 643: delete “although” and insert “although” after “,”.

l. 647: delete “,” after “Although”.

l. 655: replace “more” with “a greater”.

l. 657: delete “,”.

l. 659: delete “,” after “thus”.

l. 660: replace “component” with “components” and delete “,” after “Since”.

l. 661: delete “,”.

l. 667: insert “,” after “crucial”.

l. 668: delete “,” and replace “effects” with “affects”.

l. 672: insert “,” after “own”.

l. 674: replace “has” with “have”.

l. 675: insert “The” before “Role” (role).

l. 677: insert “a” after “contributed”.

l. 681: delete “,”.

l. 692: replace “trait” with “traits”.

l. 698: delete “,”.

l. 700: insert “,” after “GA”.

l. 704: insert “,” after “trait” and delete “an”.

l. 707: insert “,” before “which” and insert “the” before “allele”.

l. 710: delete “a”.

l. 713: insert “be” after “to” and delete “the”.

l. 714: replace “program” with “programs”.

l. 715: delete “~”.

l. 721: delete “an”.

7. PLOS authors have the option to publish the peer review history of their article (what does this mean?). If published, this will include your full peer review and any attached files.

Reviewer #1: No

---

## [Author Response · Author response to Decision Letter 1]

23 Jul 2019

Authors’ response: We HIGHLY appreciate the reviewer for thoroughly going through our manuscript and for suggesting all the corrections. We have gone through the whole manuscript again and made all the necessary corrections in the grammar (using track changes).

---

## [Decision Letter · Decision Letter 2]

16 Aug 2019

Genome wide genetic dissection of wheat quality and yield related traits and their relationship with grain shape and size traits in an elite × non-adapted bread wheat cross

PONE-D-19-14666R2

Dear Dr. Kumar,

We are pleased to inform you that your manuscript has been judged scientifically suitable for publication and will be formally accepted for publication once it complies with all outstanding technical requirements.

With kind regards,

Aimin Zhang, Ph.D.

Academic Editor

PLOS ONE

Additional Editor Comments (optional):

Reviewers' comments:

Reviewer's Responses to Questions

**Comments to the Author**

1. If the authors have adequately addressed your comments raised in a previous round of review and you feel that this manuscript is now acceptable for publication, you may indicate that here to bypass the “Comments to the Author” section, enter your conflict of interest statement in the “Confidential to Editor” section, and submit your "Accept" recommendation.

Reviewer #1: All comments have been addressed

2. Is the manuscript technically sound, and do the data support the conclusions?

Reviewer #1: Yes

3. Has the statistical analysis been performed appropriately and rigorously? 

Reviewer #1: Yes

4. Have the authors made all data underlying the findings in their manuscript fully available?

Reviewer #1: Yes

5. Is the manuscript presented in an intelligible fashion and written in standard English?

Reviewer #1: Yes

6. Review Comments to the Author

Reviewer #1: The authors have gone to great lengths to amend the manuscript based upon the suggestions made, which is much appreciated.

I therefore have no further suggestions and wish the authors all the best of luck in their future studies, which I am sure will be very fruitful.

7. PLOS authors have the option to publish the peer review history of their article (what does this mean?). If published, this will include your full peer review and any attached files.

Reviewer #1: No

---

## [Editor Report · Acceptance letter]

21 Aug 2019

PONE-D-19-14666R2 

Genome wide genetic dissection of wheat quality and yield related traits and their relationship with grain shape and size traits in an elite × non-adapted bread wheat cross 

Dear Dr. Kumar:

I am pleased to inform you that your manuscript has been deemed suitable for publication in PLOS ONE. Congratulations! Your manuscript is now with our production department. 

With kind regards,

on behalf of

Prof. Aimin Zhang 

Academic Editor

PLOS ONE